# BAD HABITS: POLICY CONFOUNDING AND OUT-OF-TRAJECTORY GENERALIZATION IN RL

## ABSTRACT

Reinforcement learning agents may sometimes develop habits that are effective only when specific policies are followed. After an initial exploration phase during which agents try out different actions in the environment, they eventually converge on a particular policy. At this point, the distribution over state-action trajectories becomes narrower, leading agents to repeatedly experience the same transitions. This repetitive exposure can give rise to spurious correlations. Agents may then pick up on these correlations and develop simple habits that only work well within the specific set of trajectories dictated by their policy. The issue here is that these habits can result in incorrect outcomes if agents are forced to deviate from their typical trajectories due to changes in the environment or in their policies. In this paper, we provide a mathematical characterization of this phenomenon, which we refer to as policy confounding, and show, through a series of examples, when and how it occurs in practice.

## 1 INTRODUCTION

> *This morning, I went to the kitchen for a coffee. When I arrived,*
> *I forgot why I was there, so I got myself a coffee—*

How often do you do something without paying close attention to your actions? Have you ever caught yourself thinking about something else while washing the dishes, making coffee, or cycling? Acting out of habit is a vital human skill as it allows us to concentrate on more important matters while carrying out routine tasks. You can commute to work while thinking about how to persuade your boss to give you a salary raise or prepare dinner while imagining your next holidays in the Alps. However, unlike in the above example, habits can also lead to undesired outcomes when we fail to recognize that the context has changed. You may hop in your car and start driving towards work even though it is a Sunday and you actually want to go to the grocery store, or you may flip the light switch when leaving a room even though the lights are already off.

Here, we show that reinforcement learning (RL) agents also struggle from this same issue. This is due to a phenomenon we term *policy confounding*, which reflects how policies, as a result of influencing both past and future observation variables, may induce spurious correlations Pearl et al. (2016) among them. These correlations can lead to the development of seemingly sensible but incorrect habits, such as automatically flipping the light switch upon leaving a room, without confirming whether the lights are actually on. The core problem here is that these habits can produce incorrect results when agents are forced to deviate from their usual trajectories due to changes in their policies or the environment. We refer to this problem as *out-of-trajectory* (OOT) generalization. It is important to note that OOT generalization differs from the standard RL generalization problem (Kirk et al., 2023) in that the objective is not to generalize to environments with different dynamics and (or) rewards but rather generalize to different trajectories within the same environment.

**Contributions** This paper introduces and characterizes the phenomenon of *policy confounding*. To do so, we provide a mathematical framework that helps us describe the different types of state representations, and reveal how, as a result of policy confounding, the agent may learn representations based on spurious correlations that do not guarantee OOT generalization. Moreover, we include a series of clarifying examples that illustrate how this occurs. Unfortunately, we do not have a complete answer for how to prevent policy confounding. However, we suggest a few off-the-shelf solutions that

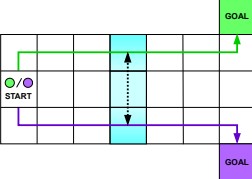 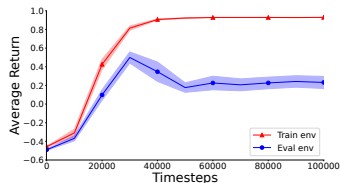

Figure 1: Left: An illustration of the Frozen T-Maze environment. Right: Learning curves when evaluated in the Frozen T-Maze environment with (blue curve) and without (red curve) ice.

may help mitigate its effects. We hope this paper will create awareness among the RL community about the risks of policy confounding and inspire further research on this topic.

## 2 EXAMPLE: FROZEN T-MAZE

We now provide an example to illustrate the phenomenon and motivate the need for careful analysis. The environment shown in Figure 1 is a variant of the popular T-Maze environment (Bakker, 2001). The agent receives a binary signal, green or purple, at the start location. Then, it needs to move to the right and reach the correct goal at the end of the maze (ignore the blue cells and the black vertical arrow in the middle of the maze for now). The agent obtains a reward of $+1$ for moving to the green (purple) goal when having received the green (purple) signal and a reward of $-1$ otherwise. There is also a $-0.1$ penalty per timestep to encourage the agent to take the shortest path toward the goal. At first sight, one may think that the only way the agent can solve the task is if, at every cell along its trajectory, it can recall the initial signal. However, once the agent figures out the shortest path to each of the two goals (depicted by the green and purple arrows), the agent may safely forget the initial signal. The agent knows that whenever it is at any of the cells along the green (purple) path, it must have received the green (purple) signal. Hence, it can simply move toward the right goal on the basis of its own location. Sticking to this habit is optimal so long as the agent commits to always taking these two paths.[1] It is also essential that the environment's dynamics remain the same since even the slightest change in the agent's trajectories may erase the spurious correlation induced by the agent's policy between the agent's location and the correct goal.

To show that this actually occurs in practice, we train agents in the original environment (train env) and evaluate them on a variant of the same (eval env), where some ice (blue) has appeared in the middle of the maze. The ice makes the agent slip from the upper cell to the bottom cell and vice versa. Note that, although the ice does change the environment dynamics, its purpose is to force the agent to take trajectories different from the optimal ones. The way we implemented it, the effect of the ice would be equivalent to forcing the agent to move down twice when in the top cell or move up twice when in the bottom cell. Importantly, these trajectories are feasible in the original environment. The plot on the right of Figure 1 shows the return averaged over 10 trials. The performance drop in the evaluation environment (blue curve) suggests that the agents' policies do not generalize. The ice confuses the agents, who, after being pushed away from their preferred trajectories, can no longer select the right goal. More details about this experiment are provided in Section 7.

## 3 RELATED WORK

The presence of spurious correlations in the training data is a well-studied problem in machine learning. These correlations often provide convenient shortcuts that a model can exploit to make predictions (Beery et al., 2018). However, the performance of a model that relies on them may significantly deteriorate under different data distributions (Quionero-Candela et al., 2009; Arjovsky, 2021). Langosco et al. (2022) show that RL agents may use certain environment features as proxies for choosing their actions. These features, which show only in the training environments, happen to be spuriously correlated with the agent's objectives. In contrast, we demonstrate that, as a result of policy confounding, agents may directly take part in the formation of spurious correlations. A few prior works have already reported empirical evidence of particular forms of policy confounding, showing that in deterministic environments, agents can rely on information that correlates with the

---

[1]Note that the two paths highlighted in Figure 1 are not the only optimal paths. However, for the agent to be able to ignore the initial signal, it is important that the paths do not overlap.

agent's progress in an episode to determine the optimal actions. This strategy is effective because under fixed policies, features such as timers (Song et al., 2020), agent's postures (Lan et al., 2023), or previous action sequences (Machado et al., 2018) can be directly mapped to the agent's state. These works provide various hypotheses to justify their experimental observations. Here, we contribute an overarching theory that explains the underlying causes and mechanisms behind these results, along with a series of examples illustrating other types of policy confounding. Please refer to Appendix C for more details on related work.

## 4 PRELIMINARIES

In this section, we introduce the notation used throughout the paper and provide the mathematical formulation of the problem.

**Definition 1** (MDP). A Markov decision process (MDP) is a tuple $\langle S, \mathcal{A}, T, R \rangle$, where $\mathcal{S}$ is the set of states, $\mathcal{A}$ is the set of actions that are available to the agent, $T : \mathcal{S} \times \mathcal{A} \rightarrow \Delta(\mathcal{S})$ is the transitions function, and $R : \mathcal{S} \times \mathcal{A} \rightarrow \mathbb{R}$ is the reward function.

In particular, we focus on problems where states are represented by a set of observation variables, or factors, (Boutilier et al., 1999). This is common when parametric functions are used to model policies and value functions (Sutton & Barto, 2018). These variables typically describe features of the agent's state in the environment.

**Definition 2** (FMDP). A Factored Markov decision process (FMDP) is an MDP where the set of states is described by a set of observation variables $\Theta = \{\Theta^1, ..., \Theta^{|\Theta|}\}$. Each variable $\Theta^i$ can take any of the values in its domain $\theta^i \in \mathrm{dom}(\Theta^i)$. Hence, every state $s$ corresponds to a different combination of values $\langle \theta^1, ..., \theta^{|\Theta|} \rangle \in \times_i \mathrm{dom}(\Theta^i) = \mathcal{S}$.

The task for the agent involves finding a policy $\pi : \mathcal{S} \rightarrow \Delta(\mathcal{A})$ that maximizes the expected discounted sum of rewards (Sutton & Barto, 2018). Yet, depending on the number of observation variables, learning a policy that conditions on all of them may be infeasible. Fortunately, in many problems, not all variables are strictly relevant; the agent can usually find compact representations of the states, that are sufficient for solving the task.

While, for simplicity, we employ the MDP formulation, the framework we introduce next is not limited to fully observable environments. In cases where the current observation does not satisfy the Markov property, meaning that current observation variables alone are insufficient for predicting state transitions, we address this limitation by augmenting $\Theta$ with past action and observation variables. In the extreme case, the agent may need to condition its decisions on the entire history, which itself is guaranteed to satisfy the Markov property (Kaelbling et al., 1998).

## 5 STATE REPRESENTATIONS

Factored representations are useful because they readily define relationships between states. States can be compared to one another by looking at the individual values the different variables take. Removing some of the variables in $\Theta$ has the effect of grouping together those states that share the same values for the remaining ones. Thus, in contrast with the classical RL framework, which treats states as independent entities, we can define state abstractions at the variable level instead of doing so at the state level (Li et al., 2006).

**Definition 3** (State representation). A state representation is a function $\Phi : \mathcal{S} \rightarrow \bar{\mathcal{S}}$, with $\mathcal{S} = \times_i \mathrm{dom}(\Theta^i)$, $\bar{\mathcal{S}} = \times_i \mathrm{dom}(\bar{\Theta}^i)$, and $\bar{\Theta} \subseteq \Theta$.

Intuitively a state representation $\Phi(s_t)$ is a context-specific projection of a state $s \in \mathcal{S} = \times_i \mathrm{dom}(\Theta^i)$ onto a lower dimensional space $\bar{s} = \times_i \mathrm{dom}(\bar{\Theta}^i)$ defined by a subset of its variables, $\bar{\Theta} \subseteq \Theta$. We use $\{s\}^{\Phi} = \{s' \in \mathcal{S} : \Phi(s') = \Phi(s)\}$ to denote the equivalence class of $s$ under $\Phi$.

### 5.1 MARKOV STATE REPRESENTATIONS

As noted in Section 4, the agent should strive for state representations with few variables. Yet, not all state representations will be sufficient to learn the optimal policy; some may exclude variables that contain useful information for the task at hand.

**Definition 4** (Markov state representation). A state representation $\Phi(s_t)$ is said to be Markov if, for all $s_t, s_{t+1} \in \mathcal{S}$, $a_t \in \mathcal{A}$,

$$R(s_t, a_t) = R(\Phi(s_t), a_t) \quad \text{and} \quad \sum_{s'_{t+1} \in \{s_{t+1}\}^\Phi} T(s'_{t+1} \mid s_t, a_t) = \Pr(\Phi(s_{t+1}) \mid \Phi(s_t), a_t),$$

where $R(\Phi(s_t), a_t) = \{R(s'_t, a_t)\}_{s'_t \in \{s_t\}^\Phi}$ is the reward at any $s'_t \in \{s_t\}^\Phi$.

The above definition is analogous to the notion of bisimulation (Dean & Givan, 1997; Givan et al., 2003) or model-irrelevance state abstraction (Li et al., 2006). Representations satisfying these conditions are guaranteed to be behaviorally equivalent to the original representation. That is, for any given policy and initial state, the expected return (i.e., cumulative reward; Sutton & Barto, 2018) is the same when conditioning on the full set of observation variables $\Theta$ or on the Markov state representation $\Phi$.

**Definition 5** (Minimal state representation). A state representation $\Phi^* : \mathcal{S} \to \bar{\mathcal{S}}^*$ with $\bar{\mathcal{S}}^* = \times_i \text{dom}(\bar{\Theta}^{*i})$ is said to be *minimal*, if all other state representations $\Phi : \mathcal{S} \to \bar{\mathcal{S}}$ with $\bar{\mathcal{S}} = \times_i \text{dom}(\bar{\Theta}^i)$ and $\bar{\Theta} \subset \bar{\Theta}^*$, for some $s \in \mathcal{S}$, are not Markov.

In other words, $\Phi^*$ is *minimal* when none of the remaining variables can be removed while the representation remains Markov. Hence, we say that a minimal state representation $\Phi^*$ is a sufficient statistic of the full set of observation variables $\Theta$.

**Definition 6** (Superfluous variable). Let $\{\bar{\Theta}^*\}_{\cup \Phi^*}$ be the union of variables in all possible minimal state representations. A variable $\Theta^i \in \Theta$ is said to be superfluous, if $\Theta^i \notin \{\bar{\Theta}^*\}_{\cup \Phi^*}$.

## 5.2 $\pi$-MARKOV STATE REPRESENTATIONS

Considering that the agent's policy will rarely visit all states, the notion of Markov state representation seems excessively strict. We now define a relaxed version that guarantees the representation to be Markov when a specific policy $\pi$ is followed.

**Definition 7** ($\pi$-Markov state representation). A state representation $\Phi^\pi(h_t)$ is said to be $\pi$-Markov if, for all $s_t, s_{t+1} \in \mathcal{S}^\pi$, $a_t \in \text{supp}(\pi(\cdot \mid s_t))$,

$$R(s_t, a_t) = R^\pi(\Phi^\pi(s_t), a_t) \quad \text{and} \quad \sum_{s'_{t+1} \in \{s_{t+1}\}^\Phi_\pi} T(s'_{t+1} \mid s_t, a_t) = \Pr^\pi(\Phi^\pi(s_{t+1}) \mid \Phi^\pi(s_t), a_t),$$

where $S^\pi \subseteq S$ denotes the set of states visited under $\pi$, $R^\pi(\Phi^\pi(s_t), a_t) = \{R(s'_t, a_t)\}_{s'_t \in \{s_t\}^\Phi_\pi}$, $\{s\}^\Phi_\pi = \{s' \in S^\pi : \Phi^\pi(s') = \Phi^\pi(s)\}$, and $\Pr^\pi$ is probability under $\pi$.

**Definition 8** ($\pi$-minimal state representation). A state representation $\Phi^{\pi*} : \mathcal{S}^\pi \to \bar{\mathcal{S}}^{\pi*}$ with $\bar{\mathcal{S}}^{\pi*} = \times_i \text{dom}(\bar{\Theta}^{\pi*i})$ is said to be $\pi$-*minimal*, if all other state representations $\Phi : \mathcal{S}^\pi \to \bar{\mathcal{S}}^\pi$ with $\bar{\mathcal{S}}^\pi = \times_i \text{dom}(\bar{\Theta}^i)$ and $\bar{\Theta} \subset \bar{\Theta}^{\pi*}$, for some $s \in \mathcal{S}^\pi$, are not $\pi$-Markov.

## 6 POLICY CONFOUNDING

We are now ready to describe how and when policy confounding occurs, as well as why we should care, and how we should go about preventing it. The proofs for all theoretical results are deferred to Appendix A.

The next result demonstrates that a $\pi$-Markov state representation $\Phi^\pi$ requires at most the same variables, and in some cases fewer, than a minimal state representation $\Phi^*$, while still satisfying the Markov conditions for those states visited under $\pi$, $s \in S^\pi$.

**Proposition 1.** *Let $\mathbf{\Phi}^*$ be the set of all possible minimal state representations, where every $\Phi^* \in \mathbf{\Phi}^*$ is defined as $\Phi^* : \mathcal{S} \to \bar{\mathcal{S}}^*$ with $\bar{\mathcal{S}}^* = \times_i \text{dom}(\bar{\Theta}^{*i})$. For all $\pi$ and all $\Phi^* \in \mathbf{\Phi}^*$, there exists a $\pi$-Markov state representation $\Phi^\pi : \mathcal{S}^\pi \to \bar{\mathcal{S}}^\pi$ with $\bar{\mathcal{S}}^\pi = \times_i \text{dom}(\bar{\Theta}^{\pi i})$ such that for all $s \in \mathcal{S}^\pi$, $\bar{\Theta}^\pi \subseteq \bar{\Theta}^*$. Moreover, there exist cases for which $\bar{\Theta}^\pi$ is a proper subset, $\bar{\Theta}^\pi \neq \bar{\Theta}^*$.*

Although the result above seems intuitive, its truth may appear incidental. While it is clear that $\Phi^\pi$ will never require more variables than the corresponding minimal state representation $\Phi^*$, whether or not $\Phi^\pi$ will require fewer, seems just an arbitrary consequence of the policy being followed.

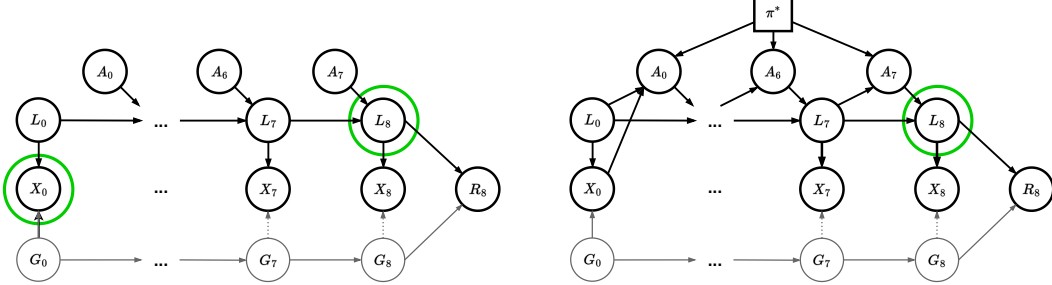

Figure 2: Two DBNs representing the dynamics of the Frozen T-Maze environment, when actions are sampled at random (left), and when they are determined by the optimal policy (right).

Moreover, since the variables in $\bar{\Theta}^*$ are all strictly relevant for predicting transitions and rewards, one may think that a policy $\pi$ inducing representations such that $\bar{\Theta}^\pi \subset \bar{\Theta}^*$ can never be optimal. However, as shown by the following example, it turns out that the states visited by a particular policy, especially if it is the optimal policy, tend to contain a lot of redundant information. This is particularly true in environments where future states are heavily influenced by past actions.

**Example 1. (Frozen T-Maze)** Let us consider the Frozen T-Maze again (Section 2). Figure 2 shows two dynamic Bayesian networks (DBN; Dean & Kanazawa, 1989; Murphy, 2002) describing the dynamics of the environment. The nodes labeled as $L$ represent the agent's location from $t = 0$ to $t = 8$. All intermediate nodes between $t = 0$ and $t = 7$ are omitted for simplicity. The nodes labeled as $G$ indicate whether the goal is to go to the green or the purple cell (see Figure 1). Note that $G$ always takes the same value at all timesteps within an episode (either green or purple). The information in $G$ is hidden and only passed to the agent at the start location through the node $X_0$. This makes the environment partially observable. Hence, states are defined as the history of actions, locations, and signal values, $s_t = \langle l_0, x_0, a_0..., a_{t-1}, l_t, x_t \rangle$. Of course, most of the information in $s_t$ is irrelevant for predicting transitions and rewards. However, depending on the policy being followed, the agent may be able to ignore more or fewer variables. This can be shown by comparing the two DBNs in Figure 2. Let us say that we want to predict the reward $R_8$ at given the state $s_8$ at time $t = 8$. On the one hand, if actions are not specified by any particular policy, but simply sampled at random (left DBN), to determine $R_8$, one needs to know the signal $X_0$ received at $t = 0$ and the agent's current location $L_8$. These are highlighted by the green circles. This is because the actions $\langle A_0, ..., A_7 \rangle$ appear as exogenous variables and can take any possible value. Hence, the reward could be either $-0.1$, (per timestep penalty), $-1$ (wrong goal), or $+1$ (correct goal) depending on the actual values of $X_0$ and $L_8$. On the other hand, when actions are sampled from the optimal policy $\pi^*$ (right DBN), knowing $L_8$ (green circle) is sufficient to determine $R_8$. In this second case, $\pi^*$ makes the action $A_0$, and thus all future agent locations, dependent on the initial signal $X_0$. This occurs because, under the optimal policy (green and purple paths in Figure 1), the agent always takes the action 'move up' when receiving the green signal or 'move down' when receiving the purple signal, and then follows the shortest path towards each of the goals. As such, we have that, from $t = 1$ onward, $\Phi^{\pi^*}(s_t) = l_t$ is a $\pi$-Markov state representation since it constitutes a sufficient statistic of the state $s_t$ under $\pi^*$. Analogously, from $t = 1$, actions may also condition only on $L$.

The phenomenon highlighted by the previous example is the result of a spurious correlation induced by the optimal policy between the initial signal $X_0$ and the agent's future locations $\langle L_1, ..., L_8 \rangle$. Generally speaking, this occurs because policies act as confounders, opening backdoor paths between future state variables $\Theta_{t+1}$ and the variables in the current state $\Theta_t$ (Pearl, 2000). This is shown by the DBN depicted in Figure 3, where we see that the policy influences both the current state variables and also future state variables, hence potentially affecting their conditional relationships. For instance, in the above example, $R^{\pi^*}(L_8 = \text{'green goal'}) = +1$ when following $\pi^*$, while for an arbitrary $\pi$, $R(L_8 = \text{'green goal'}) = \pm 1$.

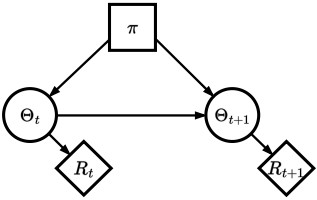

Figure 3: A DBN illustrating the phenomenon of policy confounding. The policy opens backdoor path that can affect conditional relations between the variables in $\Theta_t$ and $\Theta_{t+1}$

**Definition 9** (Policy Confounding). A state representation $\Phi : \mathcal{S} \to \bar{\mathcal{S}}$ is said to be confounded by a policy $\pi$ if, for some $s_t, s_{t+1} \in \mathcal{S}, a_t \in \mathcal{A}$,

$$R^\pi(\Phi(s_t), a_t) \neq R^\pi(\text{do}(\Phi(s_t)), a_t) \quad \text{or} \quad \text{Pr}^\pi(\Phi(s_{t+1}) \mid \Phi(s_t), a_t) \neq \text{Pr}^\pi(\Phi(s_{t+1}) \mid \text{do}(\Phi(s_t)), a_t)$$

The operator $\text{do}(\cdot)$ is known as the do-operator, and it is used to represent physical interventions in a system (Pearl, 2000). These interventions are meant to distinguish cause-effect relations from mere statistical associations. In our case, $\text{do}(\Phi(s_t))$ means setting the variables forming the state representation $\Phi(s_t)$ to a particular value and considering all possible states in the equivalence class, $s_t' \in \{s_t\}^\Phi$. That is, independently of what policy is being followed.

It is easy to show that the underlying reason why a $\pi$-Markov state representation may require fewer variables than the minimal state representation (as in Example 1) is indeed policy confounding.

**Theorem 1.** *Let $\Phi^* : \mathcal{S} \to \bar{\mathcal{S}}^*$ with $\bar{\mathcal{S}}^* = \times_i \text{dom}(\bar{\Theta}^{*i})$ be a minimal state representation. If, for some $\pi$, there is a $\pi$-Markov state representation $\Phi^\pi : \mathcal{S}^\pi \to \bar{\mathcal{S}}^\pi$ with $\bar{\mathcal{S}}^\pi = \times_i \text{dom}(\bar{\Theta}^{\pi i})$, such that $\bar{\Theta}^\pi \subset \bar{\Theta}^*$ for some $s \in \mathcal{S}$, then $\Phi^\pi$ is confounded by policy $\pi$.*

Finally, it is worth noting that even though, in Example 1, the variables included in the $\pi$-minimal state representation are a subset of the variables in the minimal state representation, $\bar{\Theta}^{\pi*} \subset \bar{\Theta}^*$, this is not always the case, as $\bar{\Theta}^{\pi*}$ may contain superfluous variables (Definition 6). An example illustrating this situation is provided in Appendix B (Example 4).

**Proposition 2.** *Let $\{\bar{\Theta}^*\}_{\cup \Phi^*}$ be the union of variables in all possible minimal state representations. There exist cases where, for some $\pi$, there is a $\pi$-minimal state representation $\Phi^{\pi*} : \mathcal{S}^\pi \to \bar{\mathcal{S}}^{\pi*}$ with $\bar{\mathcal{S}}^{\pi*} = \times_i \text{dom}(\bar{\Theta}^{\pi*i})$ such that $\bar{\Theta}^{\pi*} \setminus \{\bar{\Theta}^*\}_{\cup \Phi^*} \neq \emptyset$.*

## 6.1 WHY SHOULD WE CARE ABOUT POLICY CONFOUNDING?

Leveraging spurious correlations to develop simple habits can be advantageous when resources such as memory, computing power, or data are limited. Agents can disregard and exclude from their representation those variables that are redundant under their policies. However, the challenge is that some of these variables may be crucial to ensure that the agent behaves correctly when the context changes. In the Frozen T-Maze example from Section 2, we observed how the agent could no longer find the correct goal when the ice pushed it away from the optimal trajectory. This is a specific case of a well-researched issue known as out-of-distribution (OOD) generalization (Quionero-Candela et al., 2009; Arjovsky, 2021). We refer to it as *out-of-trajectory* (OOT) generalization to highlight that the problem arises due to repeatedly sampling from the same policy and thus following the same trajectories. In contrast to previous works (Kirk et al., 2023) that address generalization to environments that differ from the training environment, our objective here is to generalize to trajectories the agent never (or only rarely) takes.

Ideally, the agent should aim to learn representations that enable it to predict future rewards and transitions even when experiencing slight variations in its trajectory. Based on Definition 4, we know that, in general, only a Markov state representation satisfies these requirements. However, computing such representations is typically intractable (Ferns et al., 2006), and thus most standard RL methods usually learn representations by maximizing an objective function that depends on the distribution of trajectories $P^b(\tau)$ visited under a behavior policy $b$ (e.g., expected return, $\mathbb{E}_{\tau \sim P^b(\tau)}[G(\tau)]$; Sutton & Barto, 2018). The problem is that $b$ may favor certain trajectories over others, which may lead to the exploitation of spurious correlations in the learned representation.

## 6.2 WHEN SHOULD WE WORRY ABOUT OOT GENERALIZATION IN PRACTICE?

The previous section highlighted the generalization failures of representations that depend on spurious correlations. Now, let us delve into the circumstances in which policy confounding is most prone to cause problems.

**Function approximation** Function approximation has enabled traditional RL methods to scale to high-dimensional problems, where storing values in lookup tables is infeasible. Using parametric functions (e.g., neural networks) to model policies and value functions, agents can learn abstractions by grouping together states if these yield the same transitions and rewards. As mentioned before,

abstractions occur naturally when states are represented by a set of variables since the functions simply need to ignore some of these variables. However, this also implies that value functions and policies are exposed to spurious correlations. If a particular variable becomes irrelevant due to policy confounding, the function may learn to ignore it and remove it from its representation (Example 1). This is in contrast to tabular representations, where, every state takes a separate entry, and even though there exist algorithms that perform state (state) abstractions in tabular settings (Andre & Russell, 2002; Givan et al., 2003), these abstractions are normally formed offline before learning (computing) the policy, hence avoiding the risk of policy confounding.

**Narrow trajectory distributions**   In practice, agents are less prone to policy confounding when the trajectory distribution $P^b(\tau)$ is broad (i.e., when $b$ encompasses a wide set of trajectories) than when it is narrow. This is because the spurious correlations present in certain trajectories are less likely to have an effect on the learned representations. On-policy methods (e.g., SARSA, Actor-Critic; Sutton & Barto, 2018) are particularly troublesome for this reason since the same policy being updated must also be used to collect the samples. Yet, even when the trajectory distribution is narrow, there is no reason why the agent should pick up on spurious correlations while its policy is still being updated. Only when the agent commits to a particular policy should we start worrying about policy confounding. At this point, lots of the same trajectories are being used for training, and the agent may *'forget'* (French, 1999) that, even though certain variables may no longer be needed to represent the states, they were important under previous policies. This generally occurs at the end of training when the agent has converged to a particular policy. However, if policy confounding occurs earlier during training, it may prevent the agent from further improving its policy (Nikishin et al., 2022; please refer to Appendix C for more details).

## 6.3    WHAT CAN WE DO TO IMPROVE OOT GENERALIZATION?

As mentioned in the introduction, we do not have a complete answer to the problem of policy confounding. Yet, here we offer a few off-the-shelf solutions that, while perhaps limited in scope, can help mitigate the problem in some situations. These solutions revolve around the idea of broadening the distribution of trajectories so as to dilute the spurious correlations introduced by certain policies.

**Off-policy methods**   We already explained in Section 6.2 that on-policy methods are particularly prone to policy confounding since they are restricted to using samples coming from the same policy. A rather obvious solution is to instead use off-policy methods, which allow using data generated from previous policies. Because the samples belong to a mixture of policies it is less likely that the model will pick up the spurious correlations present on specific trajectories. However, as we shall see in the experiments, this alternative works only when replay buffers are large enough. This is because standard replay buffers are implemented as queues, and hence the first experiences coming in are the first being removed. This implies that a replay buffer that is too small will contain samples coming from few and very similar policies. Since there is a limit on how large replay buffers are allowed to be, future research could explore other, more sophisticated, ways of deciding what samples to store and which ones to remove (Schaul et al., 2016).

**Exploration and domain randomization**   When allowed, exploration may mitigate the effects of policy confounding and prevent agents from overfitting their preferred trajectories. Exploration strategies have already been used for the purpose of generalization; to guarantee robustness to perturbations in the environment dynamics (Eysenbach & Levine, 2022), or to boost generalization to unseen environments (Jiang et al., 2022). The goal for us is to remove, to the extent possible, the spurious correlations introduced by the current policy. Unfortunately, though, exploration is not always without cost. Safety-critical applications require the agent to stay within certain boundaries (Altman, 1999; García & Fernández, 2015). When training on a simulator, an alternative to exploration is domain randomization (Tobin et al., 2017; Peng et al., 2018; Machado et al., 2018). The empirical results reported in the next section suggest that agents become less susceptible to policy confounding when adding enough stochasticity to the environment or to the policy. Yet, there is a limit on how much noise can be added to the environment or the policy without altering the optimal policy ( Sutton & Barto, 2018, Example 6.6: Cliff Walking).

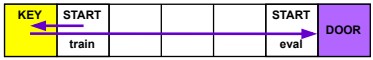 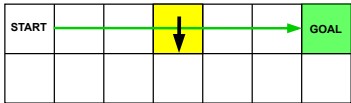

Figure 4: Illustrations of the Key2Door (left) and Diversion (right) environments.

## 7 EXPERIMENTS

The goal of the experiments is to: (1) demonstrate that the phenomenon of policy confounding described by the theory does occur in practice, (2) uncover the circumstances under which agents are most likely to suffer the effects of policy confounding and fail to generalize, and (3) evaluate how effective the strategies proposed in the previous section are in mitigating these effects.

### 7.1 EXPERIMENTAL SETUP

Agents are trained with an off-policy method, DQN (Mnih et al., 2015) and an on-policy method, PPO (Schulman et al., 2017). We represent policies and value functions as feedforward neural networks and use a stack of past observations as input in the environments that require memory. We report the mean return as a function of the number of training steps. Training is interleaved with periodic evaluations on the original environments and variants thereof used for validation. The results are averaged over 10 random seeds. Please refer to Appendix F for more details about the setup.

### 7.2 ENVIRONMENTS

We ran our experiments on three grid-world environments: the **Frozen T-Maze** from Section 2, and the below described **Key2Door**, and **Diversion** environments. We use these as pedagogical examples to clarify the ideas introduced by the theory. Nonetheless, in Appendix C, we refer to previous works showing evidence of particular forms of policy confounding in high dimensional domains.

**Example 2. Key2Door.** Here, the agent needs to collect a key placed at the beginning of the corridor in Figure 4 (left) and then open the door at the end. The current observation variables do not show whether the key has already been collected. The states are thus given by the history of past locations $s_t = \langle l_0, ..., l_t \rangle$. This is because to solve the task in the minimum number of steps, the agent must remember that it already got the key when going to the door. Yet, since during training, the agent always starts the episode at the first cell from the left, when moving towards the door, the agent can forget about the key (i.e., ignore past locations) once it has reached the third cell. As in the Frozen T-Maze example, the agent can build the habit of using its own location to tell whether it has or has not got the key yet. This, can only occur when the agent consistently follows the optimal policy, depicted by the purple arrow. Otherwise, if the agent moves randomly through the corridor, it is impossible to tell whether the key has or has not been collected. In contrast, in the evaluation environment, the agent always starts at the second to last cell, this confuses the agent, which is used to already having the key by the time it reaches said cell. A DBN is provided in Appendix D.

**Example 3. Diversion.** Here, the agent must move from the start state to the goal state in Figure 4 (right). The observations are length-8 binary vectors. The first 7 elements indicate the column where the agent is located. The last element indicates the row. This environment aims to show that policy confounding can occur not only when the environment is partially observable, as was the case in the previous examples, but also in fully observable scenarios. After the agent learns the optimal trajectory depicted by the green arrow, it can disregard the last element in the observation vector. This is because, if the agent does not deviate, the bottom row is never visited. Rather than forgetting past information, the agent ignores the last element in the current observation vector for being irrelevant when following the optimal trajectory. We train the agent in the original environment and evaluate it in a version with a yellow diversion sign in the middle of the maze that forces the agent to move to the bottom row. A DBN is provided in Appendix D.

Note that, as we did in the Frozen T-Maze environment, in order to assess whether the learned state representations generalize beyond the agent's usual trajectories we must modify the environment dynamics so as to force the agents to take alternative trajectories. Importantly, these alternative trajectories are both possible and probable in the original environments, and thus one would expect well-trained agents to perform well on them.

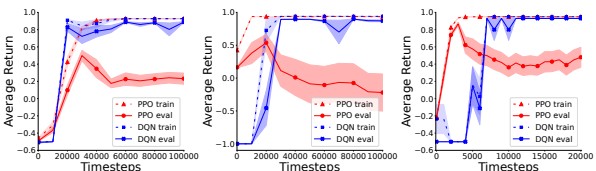 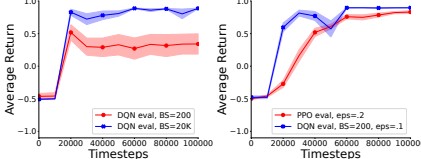

Figure 5: DQN vs. PPO in the train and evaluation variants of Frozen T-Maze (left), Key2Door (middle), and Diversion (right).

Figure 6: Frozen T-Maze. Left: DQN small vs. large buffer sizes. Right: PPO and DQN when adding stochasticity.

## 7.3 RESULTS

**On-policy vs. off-policy** The results in Figure 5 reveal the same pattern in all three environments. PPO fails to generalize outside the agent's preferred trajectories. After an initial phase where the average returns on the training and evaluation environments increase ('PPO train' and 'PPO eval'), the return on the evaluation environments ('PPO eval') starts decreasing when the agent commits to a particular trajectory, as a result of policy confounding. In contrast, since the training samples come from a mixture of policies, DQN performs optimally in both variants of the environments ('DQN train' and 'DQN eval') long after converging to the optimal policy.[2] A visualization of the state representations learned with PPO, showing that the policy does ignore variables that are necessary for generalization, is provided in Appendix E.1.

**Large vs. small replay buffers** We mentioned in Section 6.3 that the effectiveness of off-policy methods against policy confounding depends on the size of the replay buffer. The results in Figure 6 (left) confirm this claim. The plot shows the performance of DQN in the Frozen T-Maze environment when the size of the replay buffer contains 100K experiences and when it only contains the last 10K experiences. We see that in the second case, the agents performance in the evaluation environment decreases (red curve left plot). This is because, after the initial exploration phase, the distribution of trajectories becomes too narrow, and the spurious correlations induced by the latest policies dominate the replay buffer. Similar results for the other two environments are provided in Appendix E.2.

**Exploration and domain randomization** The last experiment shows that if sufficient exploration is allowed, DQN may still generalize to different trajectories, even when using small replay buffers (blue curve right plot Figure 6). In the original configuration, the exploration rate $\epsilon$ for DQN starts at $\epsilon = 1$ and decays linearly to $\epsilon = 0.0$ after 20K steps. For this experiment, we set the final exploration rate $\epsilon = 0.1$. In contrast, since exploration in PPO is normally controlled by the entropy bonus, which makes it hard to ensure fixed exploration rates, we add noise to the environment instead. The red curve in Figure 6 (right) shows that when we train in an environment where the agent's actions are overridden by a random action with $20\%$ probability, the performance of PPO in the evaluation environment does not degrade after the agent has converged to the optimal policy. This suggests that the added noise prevents the samples containing spurious correlations from dominating the training batches. However, it may also happen that random noise is not sufficient to remove the spurious correlations. As shown in Figure 13 (Appendix E.2), in the Key2Door environment, neither forcing the agent to take random actions $20\%$ of the time nor setting $\epsilon = 0.1$, solves the OOT generalization problem. Similar results for Diversion are provided in Appendix E.2.

## 8 CONCLUSION

This paper described the phenomenon of policy confounding. We showed both theoretically and empirically how as a result of following certain trajectories, agents may pick up on spurious correlations, and build habits that are not robust to trajectory deviations. We also uncovered the circumstances under which policy confounding is most likely to occur in practice and suggested a few ad hoc solutions that may mitigate its effects. We conceive this paper as a stepping stone to explore more sophisticated solutions. An interesting avenue for future research is the integration of tools from the field of causal inference (Pearl et al., 2016; Peters et al., 2017) to aid the agent in forming state representations that are grounded on causal relationships rather than mere statistical associations (Lu et al., 2018; Zhang et al., 2020; Sontakke et al., 2021; Saengkyongam et al., 2023).

---

[2]The small gap between 'DQN train' and 'DQN eval' is due to the $-0.1$ penalty per timestep. In all three environments, the shortest path is longer in the evaluation environment than in the training environment.

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

## A  PROOFS

**Lemma 1.** *Let $\mathbf{\Phi}^{\pi_1*}$ be the set of all possible $\pi$-minimal state representations under $\pi_1$, where every $\Phi^{\pi_1*} \in \mathbf{\Phi}^{\pi_1*}$ is defined as $\Phi^{\pi_1*} : \mathcal{S}^{\pi_1} \to \bar{\mathcal{S}}^{\pi_1*}$ and $\bar{\mathcal{S}}^{\pi_1*} = \times_i \mathrm{dom}(\bar{\Theta}^{\pi_1*i})$, and let $\pi_2$ be a second policy such that for all $s_t \in \mathcal{S}^{\pi_1} \cap \mathcal{S}^{\pi_2}$,*

$$\mathrm{supp}\left(\pi_2(\cdot \mid s_t)\right) \subseteq \mathrm{supp}\left(\pi_1(\cdot \mid s_t)\right).$$

*For all $\Phi^{\pi_1*} \in \mathbf{\Phi}^{\pi_1*}$, there exists a $\pi$-Markov state representation under policy $\pi_2$, $\Phi^{\pi_2} : \mathcal{S}^{\pi_2} \to \bar{\mathcal{S}}^{\pi_2}$ with $\bar{\mathcal{S}}^{\pi_2} = \times_i \mathrm{dom}(\bar{\Theta}^{\pi_2 i})$, such that $\bar{\Theta}^{\pi_2} \subseteq \bar{\Theta}^{\pi_1*}$ for all $s_t \in \mathcal{S}^{\pi_1} \cap \mathcal{S}^{\pi_2}$. Moreover, there exist cases where $\bar{\Theta}_t^{\pi_2} \neq \bar{\Theta}_t^{\pi_1*}$.*

*Proof.* First, it is easy to show that

$$\forall s_t \in \mathcal{S}, \mathrm{supp}\left(\pi_2(\cdot \mid s_t)\right) \subseteq \mathrm{supp}\left(\pi_1(\cdot \mid s_t)\right) \iff \mathcal{S}^{\pi_2} \subseteq \mathcal{S}^{\pi_1},$$

and

$$\forall s_t \in \mathcal{S}, \mathrm{supp}\left(\pi_2(\cdot \mid s_t)\right) = \mathrm{supp}\left(\pi_1(\cdot \mid s_t)\right) \iff \mathcal{S}^{\pi_2} = \mathcal{S}^{\pi_1}.$$

In particular, $\mathcal{S}^{\pi_2} \subset \mathcal{S}^{\pi_1}$ if there is at least one state $s_t' \in \mathcal{S}^{\pi_1} \cap \mathcal{S}^{\pi_2}$ such that

$$\mathrm{supp}\left(\pi_2(\cdot \mid s_t')\right) \subset \mathrm{supp}\left(\pi_1(\cdot \mid s_t')\right)$$

while

$$\mathrm{supp}\left(\pi_2(\cdot \mid s_t)\right) = \mathrm{supp}\left(\pi_1(\cdot \mid s_t)\right)$$

for all other $s_t \in \mathcal{S}^{\pi_1} \cap \mathcal{S}^{\pi_2}$.

In such cases, we know that there is at least one action $a'$ for which $\pi_2(a_t' \mid s_t') = 0$ but $\pi_1(a_t' \mid s_t') \neq 0$. Hence, if there was a state (or group of states) that could only be reached by taking action $a_t'$ at $s_t'$, $\pi_2$ would never visit it and thus $\mathcal{S}^{\pi_2} \subset \mathcal{S}^{\pi_1}$.

Further, if $\mathcal{S}^{\pi_2} \subset \mathcal{S}^{\pi_1}$, we know that, for every $\Phi^{\pi_1*} \in \mathbf{\Phi}^{\pi_1*}$, there must be a $\Phi^{\pi_2*}$ that requires, at most, the same number of variables, $\bar{\Theta}_t^{\pi_2} \subseteq \bar{\Theta}_t^{\pi_1*}$ and, in some cases, fewer, $\bar{\Theta}_t^{\pi_1*} \neq \bar{\Theta}_t^{\pi_2*}$ (e.g., Frozen T-Maze example).

$\square$

**Proposition 1.** *Let $\mathbf{\Phi}^*$ be the set of all possible minimal state representations, where every $\Phi^* \in \mathbf{\Phi}^*$ is defined as $\Phi^* : \mathcal{S} \to \bar{\mathcal{S}}^*$ with $\bar{\mathcal{S}}^* = \times_i \mathrm{dom}(\bar{\Theta}^{*i})$. For all $\pi$ and all $\Phi^* \in \mathbf{\Phi}^*$, there exists a $\pi$-Markov state representation $\Phi^\pi : \mathcal{S}^\pi \to \bar{\mathcal{S}}^\pi$ with $\bar{\mathcal{S}}^\pi = \times_i \mathrm{dom}(\bar{\Theta}^{\pi i})$ such that for all $s \in \mathcal{S}^\pi$, $\bar{\Theta}^\pi \subseteq \bar{\Theta}^*$. Moreover, there exist cases for which $\bar{\Theta}^\pi$ is a proper subset, $\bar{\Theta}^\pi \neq \bar{\Theta}^*$.*

*Proof.* The proof follows from Lemma 1. We know that, in general, $\mathcal{S}^\pi \subseteq \mathcal{S}$, and if $\pi(a_t' \mid s_t') = 0$ for at least one pair $a_t' \in \mathcal{A}, s_t' \in \mathcal{S}$ for which there is a state (or group of states) that can only be reached by taking action $a_t'$ at $s_t'$, then $\mathcal{S}^\pi \subset \mathcal{S}$. Hence, for every $\Phi^*$ there is a $\Phi^\pi$ such that $\bar{\Theta}^\pi \subseteq \bar{\Theta}^*$, and in some cases, we may have $\bar{\Theta}^\pi \neq \bar{\Theta}^*$ (e.g., Frozen T-Maze example).

$\square$

**Theorem 1.** *Let $\Phi^* : \mathcal{S} \to \bar{\mathcal{S}}^*$ with $\bar{\mathcal{S}}^* = \times_i \mathrm{dom}(\bar{\Theta}^{*i})$ be a minimal state representation. If, for some $\pi$, there is a $\pi$-Markov state representation $\Phi^\pi : \mathcal{S}^\pi \to \bar{\mathcal{S}}^\pi$ with $\bar{\mathcal{S}}^\pi = \times_i \mathrm{dom}(\bar{\Theta}^{\pi i})$, such that $\bar{\Theta}^\pi \subset \bar{\Theta}^*$ for some $s \in \mathcal{S}$, then $\Phi^\pi$ is confounded by policy $\pi$.*

*Proof.* Proof by contradiction. Let us assume that $\bar{\Theta}^\pi \subset \bar{\Theta}^*$, and yet there is no policy confounding. I.e., for all $s_t, s_{t+1} \in \mathcal{S}, a_t \in \mathcal{A}$,

$$R^\pi(\Phi^\pi(s_t), a_t) = R^\pi(\mathrm{do}(\Phi^\pi(s_t)), a_t) \tag{1}$$

and

$$\mathrm{Pr}^\pi(\Phi^\pi(s_{t+1}) \mid \Phi^\pi(s_t), a_t) = \mathrm{Pr}^\pi(\Phi^\pi(s_{t+1}) \mid \mathrm{do}(\Phi^\pi(s_t)), a_t) \tag{2}$$

First, note that the do-operator implies that the equality must hold for *all* $s_t'$ in the equivalence of $s_t$ class under $\Phi^\pi$, $s_t' \in \{s_t\}^{\Phi^\pi} = \{h_t' \in H_t : \Phi(h_t') = \Phi(h_t)\}$, i.e., not just those $h_t'$ that are visited under $\pi$,

$$R^\pi(\Phi^\pi(s_t), a_t) = R^\pi(\mathrm{do}(\Phi^\pi(s_t)), a_t) = \{R(s_t', a_t)\}_{s_t' \in \{s_t\}^\Phi} \tag{3}$$

which is precisely the first condition in Definition 4,

$$R(\Phi^\pi(s_t), a_t) = R(s_t, a_t), \tag{4}$$

for all $s_t \in \mathcal{S}$ and $a_t \in \mathcal{A}$.

Analogously, we have that,

$$\begin{aligned}
\mathrm{Pr}^\pi(\Phi^\pi(s_{t+1}) \mid \Phi^\pi(s_t), a_t) &= \mathrm{Pr}^\pi(\Phi^\pi(s_{t+1}) \mid \mathrm{do}(\Phi^\pi(s_t)), a_t) \\
&= \mathrm{Pr}(\Phi^\pi(s_{t+1}) \mid \Phi^\pi(s_t), a_t)
\end{aligned} \tag{5}$$

where the second equality reflects that the above must hold independently of $\pi$. Hence, we have that for all $s_t, s_{t+1} \in \mathcal{S}$ and $s_t' \in \{s_t\}^\Phi$,

$$\mathrm{Pr}(\Phi^\pi(s_{t+1}) \mid \Phi^\pi(s_t), a_t) = \mathrm{Pr}(\Phi^\pi(s_{t+1}) \mid \Phi^\pi(s_t'), a_t), \tag{6}$$

which means that, for all $s_t, s_{t+1} \in \mathcal{S}$ and $s_t \in \mathcal{A}$,

$$\begin{aligned}
\mathrm{Pr}(\Phi^\pi(s_{t+1}) \mid \Phi^\pi(s_t), a_t) &= \mathrm{Pr}(\Phi^\pi(s_{t+1}) \mid s_t, a_t) \\
&= \sum_{s_{t+1}' \in \{s_{t+1}\}^{\Phi^\pi}} T(s_{t+1}' \mid s_t, a_t),
\end{aligned} \tag{7}$$

which is the second condition in Definition 4.

Equations equation 4 and equation 7 reveal that if the assumption is true (i.e., $\Phi^\pi$ is not confounded by the policy), then $\Phi^\pi$ is not just $\pi$-Markov but actually strictly Markov (Definition 4). However, we know that $\Phi^*(s_t)$ is the minimal state representation, which contradicts the above statement, since, according to Definition 5, there is no proper subset of $\bar{\Theta}^*$, for all $s_t \in \mathcal{S}$, such that the representation remains Markov. Hence, $\bar{\Theta}^\pi \subset \bar{\Theta}^*$ implies policy confounding. $\qquad\square$

**Proposition 2.** *Let $\{\bar{\Theta}^*\}_{\cup \Phi^*}$ be the union of variables in all possible minimal state representations. There exist cases where, for some $\pi$, there is a $\pi$-minimal state representation $\Phi^{\pi*} : \mathcal{S}^\pi \to \bar{\mathcal{S}}^{\pi*}$ with $\bar{\mathcal{S}}^{\pi*} = \times_i \mathrm{dom}(\bar{\Theta}^{\pi*i})$ such that $\bar{\Theta}^{\pi*} \setminus \{\bar{\Theta}^*\}_{\cup \Phi^*} \neq \emptyset$.*

*Proof (sketch).* Consider a deterministic MDP with a deterministic policy. Imagine there exists a variable $X$ that is perfectly correlated with the episode's timestep $t$, but that is generally irrelevant to the task. The variable $X$ would constitute in itself a valid $\pi$-Markov state representation since it can be used to determine transitions and rewards so long as a deterministic policy is followed. At the same time, $X$ would not enter the minimal Markov state representation because it is useless under stochastic policies. Example 4 below illustrates this situation. $\qquad\square$

## B  EXAMPLE: WATCH THE TIME

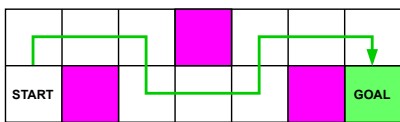

Figure 7: An illustration of the watch-the-time environment.

**Example 4. (Watch the Time)** This example is inspired by the empirical results of Song et al. (2020). Figure 7 shows a grid world environment., The agent must go from the start cell to the goal cell. The agent must avoid the pink cells; stepping on those yields a $-0.1$ penalty. There is a is $+1$ reward for reaching the goal. The agent can observe its own location within the maze $X$ and the current timestep $t$. The two diagrams in Figure 8 are DBNs describing the environment dynamics. When actions are considered exogenous random variables (left diagram), the only way to estimate the reward at $t = 10$ is by looking at the agent's location. In contrast, when actions are determined by the policy (right diagram), $t$ becomes a proxy for the agent's location $X_{10}$. This is because the start location and the sequence of actions are fixed. This implies that $t$ is a perfectly valid $\pi$-Markov state representation under $\pi^*$. Moreover, as shown by the DBN on the right, the optimal policy may simply rely on $t$ to determine the optimal action.

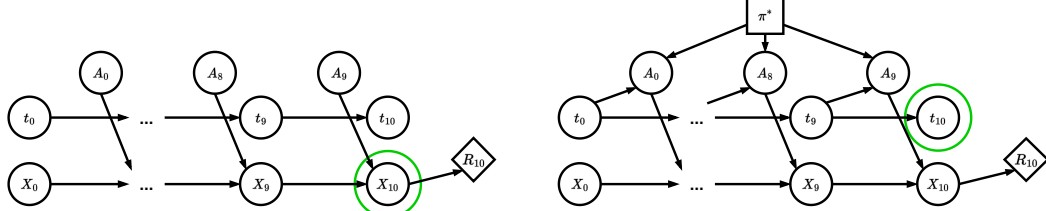

Figure 8: Two DBNs representing the dynamics of the watch-the-time environment, when actions are sampled at random (left), and when they are determined by the optimal policy (right).

## C   FURTHER RELATED WORK

**Early evidence of policy confounding**   Although to the best of our knowledge, we are the first to bring forward and describe mathematically the idea of policy confounding, a few prior works have reported evidence of particular forms of policy confounding. In their review of the Arcade Learning Environment (ALE; Bellemare et al., 2013), Machado et al. (2018) explain that because the games are fully deterministic (i.e., initial states are fixed and transitions are deterministic), open-loop policies that memorize good action sequences can achieve high scores in ALE. Clearly, this can only occur if the policies themselves are also deterministic. In such cases, policies, acting as confounders, induce a spurious correlation between the past action sequences and the environment states. Similarly, Song et al. (2020) showed, by means of saliency maps, how agents may learn to use irrelevant features of the environment that happen to be correlated with the agent's progress, such as background clouds or the game timer, as clues for outputting optimal actions. In this case, the policy is again a confounder for all these, a priori irrelevant, features. Zhang et al. (2018b) provide empirical results showing how large neural networks may overfit their training environments and, even when trained on a collection of procedurally generated environments, memorize the optimal action for each observation. Zhang et al. (2018a) shows how, when trained on a small subset of trajectories, agents fail to generalize to a set of test trajectories generated by the same simulator. Lan et al. (2023) report evidence of well-trained agents failing to perform well on Mujoco environments when starting from trajectories (states) that are out of the distribution induced by the agent's policy. We conceive this as a simple form of policy confounding. Since the Mujoco environments are also deterministic, agents following a fixed policy can memorize the best actions to take for each state instantiation, potentially relying on superfluous features. Hence, they can overfit to unnatural postures that would not occur under different policies. Finally, Nikishin et al. (2022) describe a phenomenon named 'primacy bias', which prevents agents trained on poor trajectories from further improving their policies. The authors show that this issue is particularly relevant when training relies heavily on early data coming from a fixed random policy. We hypothesize that one of the causes for this is also policy confounding. The random policy may induce spurious correlations that lead to the formation of rigid state (state) representations that are hard to recover from.

**Generalization**   Generalization is a hot topic in machine learning. The promise of a model performing well in contexts other than those encountered during training is undoubtedly appealing. In the realm of reinforcement learning, the majority of research focuses on generalization to environments that, despite sharing a similar structure, differ somewhat from the training environment (Kirk et al., 2023). These differences range from small variations in the transition dynamics (e.g., sim-to-real transfer; Higgins et al., 2017; Tobin et al., 2017; Peng et al., 2018; Zhao et al., 2020), changes in the observations (i.e., modifying irrelevant information, such as noise: Mandlekar et al., 2017; Ornia et al., 2022, or background variables: Zhang et al., 2020; Stone et al., 2021), to alterations in the reward function, resulting in different goals or tasks (Taylor & Stone, 2009; Lazaric, 2012; Muller-Brockhausen et al., 2021). Instead, we focus on the problem of OOT generalization. We aim to ensure that agents perform effectively when confronted with situations that differ from those encountered along their usual trajectories. Note that, in our experiments agents are evaluated in altered environments with different dynamics than those seen during training. These alterations are only intended to force the agent to take different trajectories. Importantly, the trajectories we force the agent to take are possible in the original environment.

**State abstraction**    State abstraction is concerned with removing from the representation all that state information that is irrelevant to the task. In contrast, we are worried about learning representations containing too little information, which can lead to state aliasing. Nonetheless, as argued by McCallum (1995), state abstraction and state aliasing are two sides of the same coin. That is why we borrowed the mathematical frameworks of state abstraction to describe the phenomenon of policy confounding. Li et al. (2006) provide a taxonomy of the types of state abstraction and how they relate to one another. Givan et al. (2003) introduce the concept of bisimulation, which is equivalent to our definition of Markov state representation (Definition 4). Ferns et al. (2006) proposes a method for measuring the similarity between two states. Castro (2020) notes that this metric is prohibitively expensive and suggests using a relaxed version that computes state similarity relative to a given policy. This is similar to our notion of $\pi$-Markov state representation (Definition 7). While the end goal of this metric is to group together states that are similar under a given policy, here we argue that this may lead to poor OOT generalization.

# D    DYNAMIC BAYESIAN NETWORKS

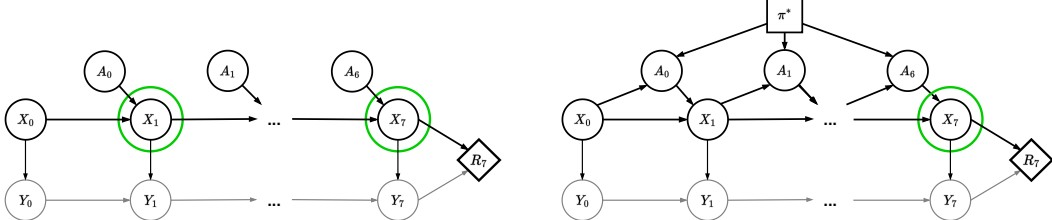

Figure 9: Two DBNs representing the dynamics of the Key2Door environment, when actions are sampled at random (left), and when they are determined by the optimal policy (right). The nodes labeled as $X$ represent the agent's location, while the nodes labeled as $Y$ represent whether or not the key has been collected. The agent can only see $X$. Hence, when actions that are sampled are random (left), the agent must remember its past locations to determine the reward $R_7$. Note that only $X_1$ and $X_7$ are highlighted in the left DBN. However, other variables in $\langle X_2, ..., X_6 \rangle$ might be needed, depending on when the key is collected. In contrast, when following the optimal policy, only $X_7$ is needed. In this second case, knowing the current location is sufficient to determine whether the key has been collected.

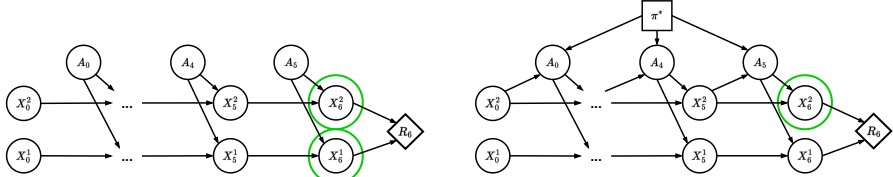

Figure 10: Two DBNs representing the dynamics of the Diversion environment, when actions are sampled at random (left), and when they are determined by the optimal policy (right). The nodes labeled as $X^1$ indicate the row where the agent is located; the nodes labeled as $X^2$ indicate the column. We see that when actions are sampled at random, both $X_6^1$ and $X_6^2$ are necessary to determine $R_6$. However, when actions are determined by the optimal policy, $X_6^2$ is sufficient, as the agent always stays at the top row.

# E    EXPERIMENTAL RESULTS

## E.1    LEARNED STATE REPRESENTATIONS

The results reported in Section 7 show that the OOT generalization problem exists. However, some may still wonder if the underlying reason is truly policy confounding. To confirm this, we compare the outputs of the policy at every state in the Frozen T-Maze when being fed the same states (observation

stack) but two different signals. That is, we permute the variable containing the signal ($X$ in the diagram of Figure 2) and leave the rest of the variables in the observation stack unchanged. We then feed the two versions to the policy network and measure the KL divergence between the two output probabilities. This metric is a proxy for how much the agent attends to the signal in every state. The heatmaps in Figure 11 show the KL divergences at various points during training (0, 10K, 30K, and 100K timesteps) when the true signal is 'green' and we replace it with 'purple'. We omit the two goal states since no actions are taken there. We see that initially (top left heatmap), the signal has very little influence on the policy (note the scale of the colormap is $10^{-6}$), after 10K steps, the agent learns that the signal is very important when at the top right state (top right heatmap). After this, we start seeing how the influence of the signal at the top right state becomes less strong (bottom left heatmap) until it eventually disappears (bottom right heatmap). In contrast, the influence of the signal at the initial state becomes more and more important, indicating that after taking the first action, the agent ignores the signal and only attends to its own location. The results for the alternative case, 'purple' signal being replaced by 'green' signal, are shown in Figure 12.

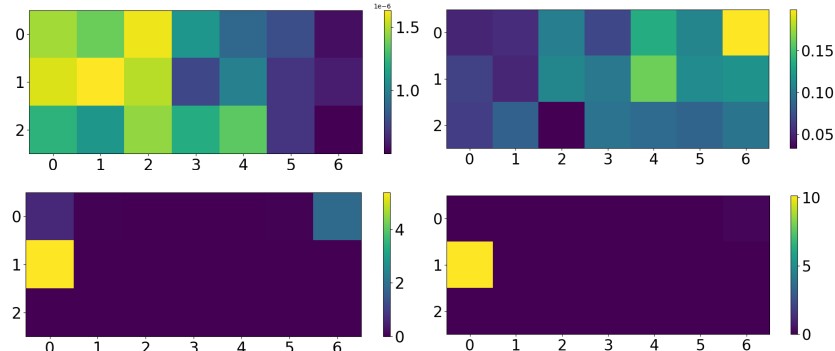

Figure 11: A visualization of the learned state representations. The heatmaps show the KL divergence between the action probabilities when feeding the policy network a stack of the past 10 observations and when feeding the same stack but with the value of the signal being switched from green to purple, after 0 (top left), 10K (top right), 30K (bottom left), and 100K (bottom right) timesteps of training.

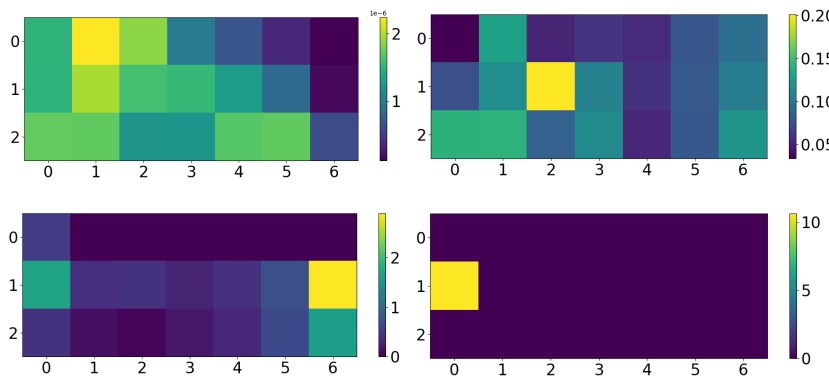

Figure 12: A visualization of the learned state representations. The heatmaps show the KL divergence between the action probabilities when feeding the policy network a stack of the past 10 observations and when feeding the same stack but with the value of the signal being switched from purple to green, after 0 (top left), 10K (top right), 30K (bottom left), and 100K (bottom right) timesteps of training.

### E.2 BUFFER SIZE AND EXPLORATION/DOMAIN RANDOMIZATION

Figures 13 and 14 report the results of the experiments described in Section 7 (paragraphs 2 and 3) for Key2Door and Diversion. We see how the buffer size also affects the performance of DQN in the

two environments (left plots). We also see that exploration/domain randomization does improve OOT generalization in Diversion but not in Key2Door.

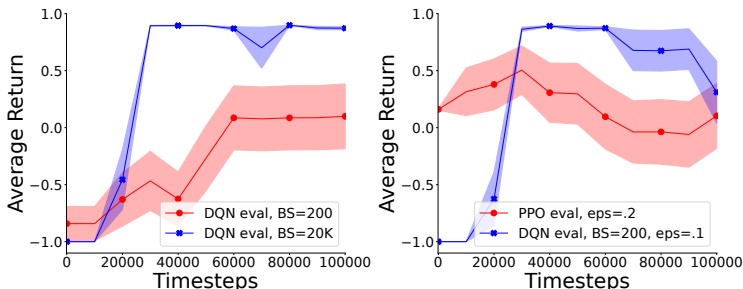

Figure 13: Key2Door. Left: DQN small vs. large buffer sizes. Right: PPO and DQN when adding stochasticity.

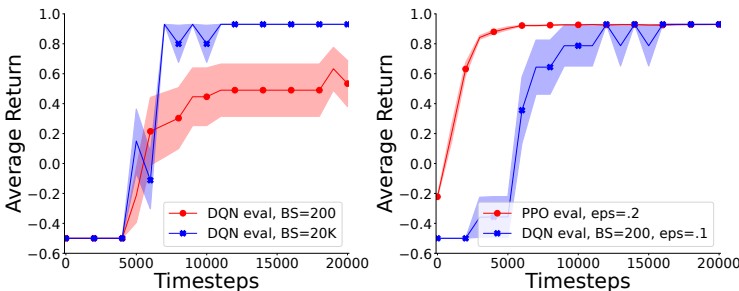

Figure 14: Diversion. Left: DQN small vs. large buffer sizes. Right: PPO and DQN when adding stochasticity.

## F   FURTHER EXPERIMENTAL DETAILS

We ran our experiments on an Intel i7-8650U CPU with 8 cores. Agents were trained with Stable Baselines3 (Raffin et al., 2021). Most hyperparameters were set to their default values except for the ones reported in Tables 1 (PPO) and 2 (DQN), which seemed to work better than the default values.

Table 1: PPO hyperparameters.

| | |
|---|---|
| Rollout steps | 128 |
| Batch size | 32 |
| Learning rate | 2.5e-4 |
| Number epoch | 3 |
| Entropy coefficient | 1.0e-2 |
| Clip range | 0.1 |
| Value coefficient | 1 |
| Number Neurons 1st layer | 128 |
| Number Neurons 2nd layer | 128 |

Table 2: DQN hyperparameters.

| | |
|---|---|
| Buffer size | 1.0e5 |
| Learning starts | 1.0e3 |
| Learning rate | 2.5e-4 |
| Batch size | 256 |
| Initial exploration bonus | 1.0 |
| Final exploration bonus | 0.0 |
| Exploration fraction | 0.2 |
| Train frequency | 5 |
| Number Neurons 1st layer | 128 |
| Number Neurons 2nd layer | 128 |

