# OpenReview forum: "Bad Habits: Policy Confounding and Out-of-Trajectory Generalization in Reinforcement Learning"
_ICLR.cc/2024/Conference — Submitted to ICLR 2024_

### Official Review · Reviewer_kJx4 · 2023-10-31

**Soundness:** 3 good
**Presentation:** 3 good
**Contribution:** 3 good
**Rating:** 5
**Confidence:** 4

**Summary:**

The paper addresses the issue of spurious correlations in reinforcement learning (RL), introducing the concept of policy confounding to formalize and analyze this problem. The authors provide theoretical insights, examples, and experimental results to demonstrate the effect of policy confounding, comparing different solutions and highlighting its impact on state representations in RL. The work aims to shed light on how an agent's policy can induce spurious correlations, potentially leading to representations that do not generalize well outside the trajectory distribution induced by the agent's policy.

**Strengths:**

The paper introduces and formalizes the concept of policy confounding, shedding light on a previously underexplored aspect of spurious correlations in RL. The theoretical framework, complemented by some examples, enhances our understanding of how policy confounding impacts state representations in RL. The manuscript is well-structured, providing a logical flow of ideas, and the writing is clear, making complex concepts accessible.

**Weaknesses:**

Clearly articulates the problem of policy confounding and relates it to issues like spurious correlations and generalization.

Provides a formal theoretical characterization grounded in state abstraction frameworks and MDPs.

Includes pedagogical examples that effectively demonstrate policy confounding.

Experiments confirm the theory and show when policy confounding is most problematic.

Highlights an important generalization challenge in RL that has been overlooked.

**Questions:**

The paper argues policy confounding poses a distinct challenge from general RL generalization, but does not extensively benchmark against recent generalization methods like invariant risk minimization, data augmentation, dynamics randomization, robust policy learning, and meta RL. These approaches could be highly relevant given the claims about out-of-trajectory generalization. While theory and simple experiments demonstrate policy confounding, how confident are you this poses a problem not already addressed by the latest techniques for improving generalization in RL? Comparisons to some of these state-of-the-art methods could better situate your claims within the broader context of research on robustness and generalization.

You demonstrate policy confounding in simple domains. Do you have evidence this manifests in more complex, high-dimensional problems? Testing on complex benchmarks could better showcase significance.

The theoretical analysis introduces useful formalisms but is very dense. For readers less familiar with this notation, could you provide more intuitive explanations of key results like Proposition 1 and Theorem 1?

While you propose some basic mitigation strategies, the paper does not offer concrete solutions. What directions seem most promising for future work to address policy confounding?

The distinction between out-of-trajectory and out-of-distribution generalization is somewhat unclear. Could you clarify this difference with explicit examples?

How does policy confounding relate to prior work on spurious correlations and generalization in RL? Are there clear differences in causes and solutions?

You cite causal representation learning as a promising direction for future work, but do not provide specifics on how these techniques could be applied to address policy confounding. Could you expand on how invariant risk minimization or other causal inference tools could help mitigate the effects you demonstrate? Are there any concrete steps you propose for integrating causal representations into solving this problem?

**Details Of Ethics Concerns:**

No ethical concerns

---

> ### Author Response · Authors · 2023-11-15
>
> We appreciate the time the reviewer invested in reviewing our work. We hope our answers below can help clear up the reviewer’s concerns.
>
> > **The paper argues policy confounding poses a distinct challenge from general RL generalization, but does not extensively benchmark against recent generalization methods like invariant risk minimization, data augmentation, dynamics randomization, robust policy learning, and meta RL. These approaches could be highly relevant given the claims about out-of-trajectory generalization.**
>
> We would like to emphasize that this is not an experimental paper. As the reviewer correctly points out, the goal of this paper is to formalize the concept of policy confounding and create awareness about its risks. We demonstrate that the issue occurs when using two of the most popular RL methods: PPO and DQN. Whether the issue manifests itself in each specific RL method is beyond the scope of this paper. Moreover, the RL methods the reviewer mentions (robust policy learning and meta RL) target other types of issues that are not directly related to policy confounding, so there is no a priori reason why these methods would help mitigate its effects. As for other generalization techniques (such as invariant risk minimization or data augmentation), it is not clear how these can be adapted to the RL setting. We do show that dynamics randomization can help mitigate its effects (last paragraph, Section 7.2).
>
> Generally, we observe the tendency in RL research to run numerous experiments and then draw conclusions based on a subset of the results. This practice is inefficient in terms of time and energy. In this paper, our approach has been to first analyze the issue analytically and then conduct targeted experiments to validate that they indeed support the proposed theory.
>
> > **While theory and simple experiments demonstrate policy confounding, how confident are you this poses a problem not already addressed by the latest techniques for improving generalization in RL? Comparisons to some of these state-of-the-art methods could better situate your claims within the broader context of research on robustness and generalization.**
>
> As mentioned in the previous response the paper highlights a new, and unexplored problem, in RL. Whether or not the latest techniques for improving generalization in RL inadvertently solve this problem does not make it less of a problem. We believe the RL community should still be warned about the risks that policy confounding poses.
>
> > **You demonstrate policy confounding in simple domains. Do you have evidence this manifests in more complex, high-dimensional problems? Testing on complex benchmarks could better showcase significance.**
>
> We did not experiment on high-dimensional domains. Our environments are meant as pedagogical examples to illustrate the problem in practice and to clarify the ideas introduced by the theory. However, the fact that standard RL methods fail already in such simple settings gives great cause for concern. Moreover, we have found plenty of empirical evidence in prior works of particular forms of policy confounding in high dimensional domains (Machado et al., 2018; Zhang et al., 2018a; Zhang et al., 2018b; Song et al., 2020; Nikishin et al., 2022; Lan et al., 2023). A review of all these works along with explanations on how the phenomena they highligt are particular instances of policy confounding is provided in Appendix C.
>
> > **The theoretical analysis introduces useful formalisms but is very dense. For readers less familiar with this notation, could you provide more intuitive explanations of key results like Proposition 1 and Theorem 1?**
>
> We acknowledge that the theory may appear somewhat intimidating. We have tried to enhance accessibility by providing numerous examples and figures. Unfortunately, defining the phenomenon mathematically requires introducing several new notations and definitions.
>
> Proposition 1 simply demonstrates that certain policies may induce state representations that require fewer observation variables than the minimal (Markov) state representation (Definition 5) while still being $\pi$-Markov (Markov for the states visited when following such policies; Definition 7). This is somewhat obvious if you consider policies that visit only a subset of the state space—think of a policy that makes the agent stay put. However, what is not so obvious is that this occurs even for well-performing (optimal) policies. The underlying reason is what we term policy confounding.  The policy introduces spurious correlations between past and future observation variables. This is illustrated by Example 1 and demonstrated by Theorem 1.

---

> ### Author Response · Authors · 2023-11-15
>
> > **What directions seem most promising for future work to address policy confounding?**
>
> As previously mentioned, our main contribution is the characterization of policy confounding. The proposed solutions, rather than serving as definitive solutions, aim to exemplify instances when the phenomenon is most likely to occur. Regarding what directions seem most promising for future work, we think that a complete reformulation of the RL problem is necesary. The agent should not only focus on maximizing rewards but should also prioritize learning robust state representations grounded in causal relations, as opposed to mere statistical associations. Hence, integrating tools from causal representation learning into the RL framework seems to be a promising path forward.
>
> > **Could you clarify the difference between OOT and OOD generalization with explicit examples?**
>
> Certainly! Out-of-trajectory (OOT) generalization is a particular instance of the more general out-of-distribution (OOD) generalization problem in RL. The objective of out-of-trajectory generalization is not to generalize to environments with different rewards and transitions (as is the case in all the previous works that target generalization in RL) but simply to alternative trajectories within the same environment.
>
> Consider a robot that is trained to go from your office to the coffee machine and back, and from your office to the printer and back. There are two different ways to go to the coffee machine: either through the printer room or through a corridor that leads directly to the coffee machine. The path through the printer room is longer, so the robot never takes it when you order coffee. However, one day you order coffee, but the corridor is blocked. Hence, the robot tries to go through the printer room and comes back with a copy of a new paper titled "Bad Habits" instead of the coffee. This is an example of out-of-trajectory generalization. Since the robot is used to getting copies when in the copy room, it ignored the order you gave it. An example of the more general problem of OOD generalization could be to ask the robot to navigate your office when the floor is wet or to order something different, such as a glass of water. The key difference is that in these two examples, the states the robot visits or the rewards it receives are different. The robot has never been trained on a wet floor, nor has it ever gotten you a glass of water before. However, in the first example, we would expect the robot to be aware that being in the copy room does not imply getting copies. To be fair, the blocked corridor represents a change in the environment; however, this change is intended to prompt the agent to choose an alternative path. Note that this alternative path was also viable in the original environment.
>
> > **How does policy confounding relate to prior work on spurious correlations and generalization in RL?**
>
> Our response to the previous comment addresses the differences concerning prior works on generalization in RL. As for works investigating spurious correlations, we are only aware of the paper by Langosco et al. (2022). The key difference in Langosco’s paper is that spurious correlations are already present in the environment. The setting is as follows: a certain object is always next to the actual goal, so the agent learns to associate the object with the goal. However, in the test environments, the object is no longer present. In our case, it is the policy itself that introduces the spurious correlations, such as always getting copies when at the printing room. As for solutions, some of the methods that work for the general RL generalization problem would likely mitigate the OOT generalization problem, such as domain randomization (Tobin et al., 2017; Peng et al., 2018; Machado et al., 2018) and exploration (Eysenbach & Levine, 2022; Jiang et al., 2022). However, we believe that coming up with more targeted strategies is crucial.
>
> > **Could you expand on how invariant risk minimization or other causal inference tools could help mitigate the effects you demonstrate? Are there any concrete steps you propose for integrating causal representations into solving this problem?**
>
> Although this is beyond the scope of this paper, we are happy to provide some hints regarding what we believe are the most promising strategies. One option would be to exploit the notion of prediction invariance (Peters et al., 2016; Arjovsky et al., 2019). Since state representations that are robust to trajectory deviations must be invariant across policies, one could enforce this property by modifying the objective function (Zhang et al., 2020). Another alternative is to actively encourage the agent to stress-test its representations in the environment by trying out alternative trajectories. This could be achieved by adding intrinsic rewards (Sontakke et al., 2021). We would be happy to include this information in the appendix if the reviewer deems it important.

---

> > ### Author Response · Authors · 2023-11-22
> >
> > Dear reviewer,
> >
> > As the discussion period comes to a close, we kindly request your feedback regarding our rebuttal. Are there any remaining concerns that are impeding the reviewer's recommendation for acceptance?

---

### Official Review · Reviewer_besR · 2023-10-31

**Soundness:** 3 good
**Presentation:** 2 fair
**Contribution:** 3 good
**Rating:** 5
**Confidence:** 4

**Summary:**

The authors investigate if RL agents develop myopic, suboptimal habits that rely on spuriously correlated inputs to make decisions. They formally define policy confounding by considering minimal representations of a policy's marginal state distribution, and then showing that these minimal representations do not generalize, like if the policy gets teleported to a state outside of it. They evaluate on toy gridworld settings where confounding happens, and show that some strategies that increase the diversity of data experienced by the agent (off-policy, exploration, etc.) reduces confounding.

**Strengths:**

The study of generalization for RL is quite important.
The study of the training dynamics of deep RL is also quite important.
This paper addresses both such problems, since policy confounding can happen both during training and evaluation.

Overall, I appreciated the theoretical definition of policy confounding, and find the way the authors choose to define it by considering the representations induced by a policy to be intuitive. However, there are several writing issues (see below) that I fear will make this paper's message miss the mark.

The experiments the author chose, while toy, highlight the policy confounding problem well. The proposed improvements, while not novel, are simple and intuitive for tackling confounding.

**Weaknesses:**

## Presentation
Overall, the paper is very awkward for reading. The authors propose many definitions and setup notations in the early pages before finally arriving to the definition of policy confounding (Sec. 6). I would suggest for the authors to reformat the paper to first give a high level sketch or intuition of the definition, perhaps aided by a visual figure, and then explaining the numerous notations and definitions required for defining policy confounding.

In section 2, the authors provide a very lengthy (almost page long) textual description of policy confounding. This is very difficult to read, I would advise the authors to move explanations into visual aids, split up into paragraphs, highlight key steps, etc.

Indeed, the authors have a tendency of writing long paragraphs and describing things step by step at a low level, rather than summarizing high level points, splitting points into multiple paragraphs, etc.  Section 6, Example 1 is almost impossible to read. I suspect most readers will not even bother reading this.

## Experimentation
The authors can improve in two ways. First, they can connect their theory more to their toy experiments. For example, is it possible to come up with a toy environment where all possible representations can be enumerated and tracked? Then it could be interesting to see what kinds of representations RL algorithms learn, and if they indeed are minimal and do not generalize.

Next, it would be interesting to investigate confounding in existing deep RL tasks, especially in well-known benchmarks (Atari, DMC, etc.), and to see if some RL algorithms are better than others. For example, would MBRL algorithms fare better or worse than model-free RL algorithms?

**Questions:**

Can the authors improve the presentation?

Can the authors think of experiments that analyze and connect to their theory more?

Can the authors showcase policy confounding in non-toy tasks in deep RL (continuous states, actions, well-known benchmark, etc.)?

---

> ### Author Response · Authors · 2023-11-15
>
> We appreciate the time the reviewer invested in reviewing our work. We hope our answers below can help clear up the reviewer’s concerns.
>
> > **Overall, the paper is very awkward for reading. The authors propose many definitions and setup notations in the early pages before finally arriving to the definition of policy confounding (Sec. 6).**
>
> We acknowledge that the theory may appear somewhat intimidating. We have tried to enhance accessibility by providing numerous examples and figures. Unfortunately, defining the phenomenon mathematically requires introducing several new notations and definitions.
>
> > **I would suggest for the authors to reformat the paper to first give a high level sketch or intuition of the definition, perhaps aided by a visual figure, and then explaining the numerous notations and definitions required for defining policy confounding.**
>
> Both the introduction and the example in Section 2 are intended to provide a high-level intuition of policy confounding. We would be happy to correct the text and add more explanations where needed if the reviewer could point out what specific parts of the text are unclear.
>
>
> > **In section 2, the authors provide a very lengthy (almost page long) textual description of policy confounding. This is very difficult to read, I would advise the authors to move explanations into visual aids, split up into paragraphs, highlight key steps, etc.**
>
> Section 2 is, in fact, an example of policy confounding in a gridworld environment. The example is accompanied by Figure 2, which depicts a sketch of the environment and a plot showing its practical effects. We kindly ask the reviewer to specify any additional visual aids they would like us to include. Regarding the text, we will split it into another paragraph beginning with "At first sight"  to enhance readability. Any other suggestions regarding which specific parts of the text are unclear would be greatly appreciated.
>
> > **The authors can improve in two ways. First, they can connect their theory more to their toy experiments. For example, is it possible to come up with a toy environment where all possible representations can be enumerated and tracked? Then it could be interesting to see what kinds of representations RL algorithms learn, and if they indeed are minimal and do not generalize.**
>
> We did analyze the internal representations learned by the RL algorithms in the Frozen T-Maze environment. The results of this experiment are reported in Appendix E. The experiment demonstrates that indeed the agent does learn a representation that does not generalize beyond the trajectories followed by the optimal policy.
>
> > **Next, it would be interesting to investigate confounding in existing deep RL tasks, especially in well-known benchmarks (Atari, DMC, etc.).**
>
> We would like to emphasize that this is not an experimental paper. Our main contribution is the characterization of the phenomenon of policy confounding. The goal of our experiments is to verify that the phenomenon described by the theory does occur in practice. To that end we have designed as set of environments, which are intended as pedagogical examples and serve to to illustrate the problem in practice and to clarify the ideas introduced by the theory.
>
> That being said, the fact that standard RL methods fail already in such simple settings gives great cause for concern. Moreover, we have found plenty of empirical evidence in prior works of particular forms of policy confounding in high dimensional domains (Machado et al., 2018; Zhang et al., 2018a; Zhang et al., 2018b; Song et al., 2020; Nikishin et al., 2022; Lan et al., 2023). A review of all these works along with explanations on how the phenomena they highligt are particular instances of policy confounding is provided in Appendix C.
>
> > **and to see if some RL algorithms are better than others. For example, would MBRL algorithms fare better or worse than model-free RL algorithms?**
>
> We do experiment with two types of RL methods: on-policy (PPO) and off-policy (DQN). The reason for this experiment is that, according to the theory, off-policy methods should be less prone to policy confounding than on-policy methods as the former train on a more diverse set of trajectories. This is confirmed by the experiments (See Figure 5). There is no reason why model-based RL (MBRL) methods would not suffer from policy confounding as much as model-free methods.
>
> Generally, we observe the tendency in RL research to run numerous experiments and then draw conclusions based on a subset of the results. This practice is inefficient in terms of time and energy. In this paper, our approach has been to first analyze the issue analytically and then conduct targeted experiments to validate that they indeed support the proposed theory.

---

> > ### Comment · Reviewer_besR · 2023-11-21
> > **Readability is still a major issue.**
> >
> > The paper is still overall, quite hard to parse for a general reader.  The promises of simply compressing text and adding paragraph breaks will not be enough to improve the readability. The paper needs to be rewritten with clear intentions of saliency for the reader.
> >
> > I highly recommend the authors ask researchers who are not in this immediate subfield, e.g. deep RL or deep learning, to read this paper and get feedback on readability.
> >
> > There are simply too many parts in this paper which suffer from readability issues for me to give low-level feedback on everything, but I will give some low-level feedback for one example, and ask the authors to apply my advice to other parts.
> >
> > ### Section 2
> > This section just "firehoses" the reader with information about the environment. The problem here is that there are a few key environmental properties that you should emphasize to the reader, but those properties get lost in the verbatim. This paragraph can be compressed, and other details moved to appendix.
> >
> > Here's how I would rewrite it:
> >
> > > In the T-Maze environment, a goal-conditioned agent is tasked with reaching either the top or bottom goal state. During training time, the environment is deterministic, allowing the agent to memorize optimal paths for reaching each goal (see Figure 1). *Because the optimal paths occupy disjoint states, the agent can deduce the optimal path through its current state rather than the goal input.*
> > >
> > >Indeed, this is a key property of policy confounding that we investigate. While seemingly innocuous, the confounding in policy inputs leads to brittle policies sensitive to distribution shift. What happens if we introduce some slippery ice in the maze?
> >
> > >Refer to the left of Figure 1. (1) The agent is tasked with going to the top goal.
> > (2) It starts off on the correct path (green), but slips over some ice and moves downwards.
> > (3) The agent finds itself in a bottom state, which it associates with going towards the bottom goal from training time.
> > (4) The agent now incorrectly goes towards the bottom goal.
> >
> > >We train agents in the deterministic environment, and test them in episodes with icy states during evaluation.  As seen in the right plot of Fig 1, the performance drop between train and test show that the policies fail to generalize to the new dynamics. See Appendix X for more details.
> >
> > ---
> >
> > Then, in Appendix, X, you can put all the details about reward functions, evaluation details, etc. No need for all of that in the main text, especially since this section is meant to be pedagogical.
> >
> > ## Suggested areas for rewriting
> > Section 6, example 1 is a huge paragraph needs to be rewritten. There are many low level details mixed in, some important, some not. Please think about what are the key environmental properties and their implications, and highlight that to the reader.
> >
> > Section 7.2, environmental descriptions also just dump all the details in a somewhat arbitrary order to the reader.
> >
> > Section 4,5,6 - It would be helpful to have a running example environment here, and then use it as an example while laying out all of the definitions. For example, if we use the T-maze, then you could give concrete examples of minimal state representations, superfluous variables, $\pi$-markov state representations, etc. This will help an outsider ground the theory to an environment they understand. I see you do refer to the Frozen T-maze in section 6, but even that is hard to read, for aforementioned reasons.

---

> ### Author Response · Authors · 2023-11-21
>
> We appreciate the input provided by the reviewer, which will help us enhance the quality of the paper. We will address these points in the next revision.
>
> However, we would like to emphasize that a rejection recommendation based solely on presentation issues seems excessive, especially considering that the quality of the presentation has been highly praised by reviewers BRdu and kJx4.
>
> Are there any remaining concerns that are impeding the reviewer's recommendation for acceptance?

---

> > ### Comment · Reviewer_besR · 2023-11-22
> > **I will retain my current score.**
> >
> > > However, we would like to emphasize that a rejection recommendation based solely on presentation issues seems excessive, especially considering that the quality of the presentation has been appraised by reviewers BRdu and kJx4.
> >
> > While I understand that this may seem excessive, the paper has failed to clearly convey its core technical ideas to me, a reviewer that has spent a decent amount of effort in understanding and reading it, adequately in its current iteration. I believe that this will happen with other readers, especially those from adjacent subfields who are interested in this phenomenon. As a result, I cannot personally recommend accept for this paper, hence my current score.

---

### Official Review · Reviewer_aFjn · 2023-11-01

**Soundness:** 2 fair
**Presentation:** 2 fair
**Contribution:** 2 fair
**Rating:** 5
**Confidence:** 3

**Summary:**

The authors discuss policy confounding, the phenomenon where RL agents rely on spurious correlations for their policies.  They define and study this problem from a theory standpoint, and empirically evaluate several proposed solutions.

**Strengths:**

This is an interesting, significant topic that is worthy of study, and I think that, issues below aside, authors have a solid approach to the problem.

**Weaknesses:**

See questions below for a couple of clarity weaknesses.

In Example 1 (Section 6), X_t and x_t seem to not be defined when they are first used.  Later, it is stated that X_0 is the signal received at t=0, but even then, it is not clear how X differs from G.  These should be better defined.

The do operator is not well-defined.  Based on the informal description (just below Definition 9), I thought that it meant that there exists an s_t, such that we consider all possible values of Phi(s_t), and adversarially pick one to try to meet one of the conditions in Definition 9 if possible (even if this is the correct interpretation, it is not clear).  However, based on a few readings for Section 6.0, I suspect the correct interpretation is something different.

It is not clear to me how exactly Example 1 corresponds to the ideas before and after it (perhaps this confusion would be resolved by resolving the confusions above).  The implication appears to be that the two different L_8 positions under the optimal policy (i.e., the green and purple positions in Figure 1) are equivalent under some Phi.  But, per Definition 3, this is not how state representations work: we can discard some subset of the state to get our state representation, but I do not see how the two different L_8 positions can have the same state representation per that definition.  So either I am misunderstanding the implication (possibly due to the clarity problems mentioned above), or else there might be a fundamental error in the example or the definitions.

Another possibility that may explain some of my confusion is that L_t and l_t do not encode the full position, but instead only the timestep or the “horizontal” position.  If this is the case, this confusion is caused by more imprecise/incomplete definitions.

Justifying/motivating the work: The examples given focus on the idea that we train on one MDP, and then evaluate on another.  This approach illustrates the issue well, but is too contrived to provide motivating examples that show why we should care about policy confounding in practice (and the experiments are all set up this way as well).  6.2 attempts to address this by discussing when we should worry in practice: function approximation and narrow trajectory distributions.  However, these two paragraphs are far too short and high-level to truly justify this work.  I believe that this is an interesting topic worthy of study, but I think this paper in its current form does a poor job of showing the reader that this is an interesting topic worthy of study.  Perhaps showing that policy confounding can be a problem in less contrived settings (for example, when the MDP or distribution of MDPs does not shift between training and evaluation) would help address this weakness.  Alternatively, another approach could be to focus a future version of the paper on a setting where the training and evaluation MDPs are inherently different, such as "sim-to-real" robotics.

The main theorem seems almost trivial, and the empirical work is based on extremely simple toy gridworlds.  So even aside from the issues above, the contribution may be a bit light.

**Questions:**

Starting at definition 2, the paper became difficult to follow.  Are \Theta^1, \Theta^2 random variables (RVs)?  For a while, I couldn’t figure out what they were (I was thinking that they were sets like \Theta, but that didn’t make complete sense, and then I was thinking that there must be typos in the definition, before I realized that they might be RVs).  If they are RVs, a statement that they are RVs, as opposed to \Theta, which is a set, would be helpful.

The \cross_i notation was extremely confusing (I almost gave up on trying to understand the paper over this).  The interpretation I settled on for \cross_i dom(\Theta^i) is dom(\Theta^1) \cross dom(\Theta^2) \cross …  Is this interpretation correct?  A definition would help clarify.

---

> ### Author Response · Authors · 2023-11-15
>
> We would like to thank the reviewer for taking the time to review our work. We believe there are certain misconceptions about the paper that we hope our responses below can clear up.
>
> > **It is not clear how Example 1 corresponds to the ideas before and after.**
>
> The reviewer seems to be confusing state (observation) variables with states and state representations. In Example 1, the sequence of loctions $L_0,... L_t$, signals $X_0,..., X_t$, goals $G_0, …, G_t$, and actions  $A_0, …, A_t$ are the state variables from which only $L_0,... L_t$, $X_0,..., X_t$, and  $A_0, …, A_t$ are observed, we call the latter the set observation variables and denote it by $\Theta_t$. States are abstract entities that can be represented by the observation variables (see Definitions 1 and 2).
>
> Now, let's clarify how the example relates to the definitions and theoretical results.  There are two scenarios in Example 1, which correspond to the two DBNs in Figure 2.
>
> In the first scenario (Figure 2, left), actions are exogenous variables and thus taken at random. Hence, the only valid state representation is $\langle X_0, L_t \rangle$ (i.e., the agent needs to condition ob its location and the signal received at the start location). This representation is Markov (Definition 4) and minimal (Definition 5).
>
> In the second scenario (Figure 2, right), actions are sampled from the optimal policy. In this case, states can be represented simply as $\langle L_t \rangle$. This representation is $\pi$-Markov (Definition 7) and $\pi$-minimal (Definition 8). The reason for this is that under the optimal policy, the agent’s location $L_t$ becomes a proxy for the signal at the start location $X_0$ ($L_t$ and $X_0$ are perfectly correlated). Hence, the agent can ignore the signal and still predict transitions and rewards. In particular, $R^{\pi^*}(L_8 = \text{green goal}) = +1$ when following $\pi^*$, while in the first scenario since $L_8$ and $X_0$ are not correlated, $R(L_8 = \text{green goal}) = \pm1$.
>
> As demonstrated by Theorem 1, and highlighted by the diagram in Figure 3, the underlying phenomenon that makes $L_t$ and $X_0$ become correlated is policy confounding.
>
> > **The do operator is not well-defined.**
>
> The do-operator is commonly used in the causal inference literature to distinguish cause-effect relationships from mere statistical associations (Pearl, 2000). We define do-operator in the text below Definition 9: $\text{do}(\Phi(s_t))$ means setting the variables forming the state representation $\Phi(s_t)$ to a particular value and considering all possible states in the equivalence class, $s'_t \in \{s_t\}^\Phi$ (i.e., all states that share the same value for the state variables that are "selected” by $\Phi$). We understand this definition may be hard to parse and will add the following example below the definition for clarification:
>
> In the T-maze environment, $R^{\pi^*}(\text{do}(L_8 = \text{green goal})) \neq R^{\pi^*}(L_8 = \text{green goal})$, since $R^{\pi^*}(\text{do}(L_8 = \text{green goal})) = \pm1$ and $R^{\pi^*}(L_8 = \text{green goal}) = +1$. This is already explained in the sentence before Definition 9. However, we will move it after the definition to make use of the $\text{do}(\cdot)$ operator.
>
> > **The examples given focus on the idea that we train on one MDP, and then evaluate on another. (...) showing that policy confounding can be a problem in less contrived settings (for example, when the MDP does not shift between training and evaluation) would help address this weakness.**
>
> As explained in the introduction the objective of OOT generalization is not to generalize to other MDPs (as is the case in the standard RL generalization problem). The objective is to generalize to different trajectories within the same environment. In order to test this we must force the agent to take alternative trajectories. The only way to do so is by modifying the MDP. However, as explained in the last paragraph of section 7.2: "these alternative trajectories are both possible and probable in the original environments, and thus one would expect well-trained agents to perform well on them."
>
> > **6.2 attempts to address when we should worry about policy confounding in practice. However, these two paragraphs are far too short and high-level to truly justify this work.**
>
> We do not intend to justify our work on these two paragraphs. These two paragraphs are meant to clarify when the phenomenon of policy confounding described by all the previous theoretical results and examples is more likely to occur in practice. They also bridge the gap between the theoretical sections and the experiments.
>
> > **The main theorem seems almost trivial.**
>
> The theorem may seem trivial after having illustrated the phenomenon with an example and having provided the definition of policy confounding. However, it is certainly not trivial that the underlying reason why in some situations agents learn representations that are not fully Markov is policy confounding.

---

> > ### Author Response · Authors · 2023-11-15
> >
> > >  **The empirical work is based on extremely simple toy gridworlds.**
> >
> > The environments are meant as pedagogical examples to illustrate the problem in practice and to clarify the ideas introduced by the theory. Our main contribution is the characterization of policy confounding.
> >
> > That being said, the fact that standard RL methods fail already in such simple settings gives great cause for concern. Moreover, we have found plenty of empirical evidence in prior works of particular forms of policy confounding in high dimensional domains (Machado et al., 2018; Zhang et al., 2018a; Zhang et al., 2018b; Song et al., 2020; Nikishin et al., 2022; Lan et al., 2023). A review of all these works along with explanations on how the phenomena they highligt are particular instances of policy confounding is provided in Appendix C.
> >
> > > **In Example 1 (Section 6), X_t and x_t seem to not be defined when they are first used. Later, it is stated that X_0 is the signal received at t=0, but even then, it is not clear how X differs from G. These should be better defined.**
> >
> > Thanks for pointing this out. The reviewer’s interpretation is correct. $X$ represents the signal which takes the same value as $G$ at the start location. At any other location, $X$ takes a dummy value. We will clarify this in the text.
> >
> > > **Are \Theta^1, \Theta^2 random variables (RVs)?**
> >
> > Yes,  $\Theta = [ \Theta^1, \Theta^2 … ]$ are random variables. As explained in Definiton 2, $\Theta = [\Theta^1, \Theta^2 … ]$ are observation variables.  The different combinations represent the different states of the environment.  We will clarify this in the revision.
> >
> > > **The \cross_i notation was extremely confusing (I almost gave up on trying to understand the paper over this). The interpretation I settled on for \cross_i dom(\Theta^i) is dom(\Theta^1) \cross dom(\Theta^2) \cross … Is this interpretation correct? A definition would help clarify.**
> >
> > This is correct. We will add the definition in the revision.

---

> > > ### Comment · Reviewer_aFjn · 2023-11-20
> > > **After Reading Authors' Response and Other Reviews**
> > >
> > > I believe that the clarifications discussed in this thread and the other review threads will help make this paper stronger and clearer.    In addition to the improvements discussed in this review's thread above, I suggest the authors spend some time working on the clarity improvements suggested by other reviewers, especially those suggested by Reviewer besR.  In anticipation of extensive rewriting and clarity improvements throughout, I am updating my score from 3 to 5.
> > >
> > > However, even assuming extensive rewriting and clarity improvements throughout, I cannot recommend an accept due to the concerns about the contribution being light, and the concerns about the experiments being too trivial or too contrived (all 3 other reviewers shared similar concerns).
> > >
> > > Suggestions: It is possible that all reviewers are misunderstanding the contribution to some extent, and that the theory contributions and the gridworld experiments are sufficiently strong contributions.  In this case, the clarity improvements should be enough for a future version of this work to be accepted.  However, since all 4 reviewers shared similar concerns, the authors may want to consider adding additional contribution in a future version.

---

> > > > ### Author Response · Authors · 2023-11-22
> > > >
> > > > We would like to thank the reviewer for responding to our rebuttal. We also think the clarifications discussed here will help improve the quality of the presentation. However, we would also like to point out that the quality of the presentation has been highly praised by reviewers BRdu and kJx4.
> > > >
> > > > > Suggestions: It is possible that all reviewers are misunderstanding the contribution to some extent, and that the theory contributions and the gridworld experiments are sufficiently strong contributions.
> > > >
> > > > We indeed feel that the reviewer has understimated the significance of our work.
> > > >
> > > > > However, since all 4 reviewers shared similar concerns, the authors may want to consider adding additional contribution in a future version.
> > > >
> > > > We would like to highlight that both reviewers BRdu and kJx4 have expressed their appreciation for the paper's current contribution.

---

### Official Review · Reviewer_BRdu · 2023-11-03

**Soundness:** 3 good
**Presentation:** 4 excellent
**Contribution:** 3 good
**Rating:** 6
**Confidence:** 4

**Summary:**

The phenomenon of "policy confounding" is identified, where an RL agent may learn to discard important information in the state due its policy focusing only on some small subset of states. Some theory is developed to understand conditions where the effect may arise and experiments in toy domains illustrate the identified phenomenon.

**Strengths:**

The phenomenon of "policy confounding" is intriguing and novel. The presentation of the paper is great, the text is easy to follow and the figures are clear. Explanations are given in sufficient detail.
There are interesting connections made to a causality perspective and the theoretical framework used (considering a factored MDP) is well-chosen. The experiments clearly demonstrate the effect in a few illustrative environments.

Overall, I find the main idea of paper to be very interesting and potentially relevant in many situations where RL agents are trained.

**Weaknesses:**

The primary weakness is that the experiments focus on toy examples specifically designed to elicit the problem. It's not entirely clear if this problem is relevant in more realistic settings. Appendix C does contain examples of failures from previous works which may be related to policy confounding though.

A slight weakness is that no mechanisms outside of conventional strategies (e.g. experience replay) are proposed to address policy confounding.

**Questions:**

I'd be willing to increase my score based on the responses to the following questions.

- T-Maze, sec2. It could be more clear to mention why the avg return is around 0.2. Is this around the level that is to be expected? Why is there a little bump around step 30k for the eval env?

- As an alternate demonstration, instead of adding the ice to the environment, why not simply place the agent directly on the state above (or below) the original starting location and see how it fares on the train env? Then, it would clear to see that if the agent always follows the trajectory it expects instead of using the color signal. We wouldn't need to introduce this additional mechanism of the ice or modify the transition function.

- About the formal definition (def 9) of policy confounding:
	It would be helpful to understand this definition by explaining how it applies to the T-maze.
	Could you clarify the meaning of the do() operator here from a mathematical point of view?
	Would it be the same as writing that the equality has to hold for all $s_t$ s.t. $\phi(s_t)$ rather than only the ones visited by $\pi$?

- About "Narrow trajectory distributions". If the environment transitons leads to a diverse set of states, would we still observe policy confounding? What if the agent is incentivized to explore through exploration bonuses? Could this avoid the issue?

- The paper makes a link between policy confounding and causality (or lack of it). Are there any tools from causality that could be used to address the problem?

-  Is policy confounding inevitable to some extent? It seems to be a consequence of having policies focus on certain parts of the state space.

---

> ### Author Response · Authors · 2023-11-15
>
> We sincerely appreciate the time the reviewer invested in reviewing our work and providing useful feedback. We are very much encouraged by the overall positive evaluation and especially by the reviewer’s acknowledgment of the significance of our work.
>
> Please find the answers to your questions below:
>
> > **The primary weakness is that the experiments focus on toy examples.**
>
> We would like to point out that the environments are meant as pedagogical examples to illustrate the problem in practice and to clarify the ideas introduced by the theory. Our main contribution is the characterization of policy confounding. Moreover, the fact that standard RL methods fail already in such simple settings gives great cause for concern.
>
> > **mention why the avg return is around 0.2. Is this around the level that is to be expected?**
>
> 0.2 is the average return over 10 runs. It is hard to expect anything from neural networks when being evaluated out-of-distribution. In some of the runs, the agent is stuck doing nothing for a while and then goes to one of the two goals. In some others, the agent simply goes to the wrong goal.
>
> > **As an alternate demonstration, instead of adding the ice to the environment, why not simply place the agent directly on the state above (or below) the original starting location and see how it fares on the train env?**
>
> We agree with the reviewer that placing the agent directly outside its usual path would make everything much simpler. Yet, some may argue that the agent has never been trained on such trajectories and, therefore, it is reasonable that it is not able to perform well. To address this type of criticism, we made sure that the trajectories the agent is evaluated on are both possible and probable in the original environments.
>
> The reviewer may find the experiment reported in Appendix E.1 interesting. In this experiment we manually permuted the value of the signal variable (X in Figure 2) to see whether the agent attends to it or not.
>
> > **About the formal definition (def 9) of policy confounding: It would be helpful to understand this definition by explaining how it applies to the T-maze.**
>
> In the T-maze environment, $R^{\pi^*}(\text{do}(L_8 = \text{green goal})) \neq R^{\pi^*}(L_8 = \text{green goal})$, since $R^{\pi^*}(\text{do}(L_8 = \text{green goal})) = \pm1$ and $R^{\pi^*}(L_8 = \text{green goal}) = +1$. This is already explained in the sentence before Definition 9. However, we will move it after the definition to make use of the $\text{do}(\cdot)$ operator.
>
> > **Could you clarify the meaning of the do() operator here from a mathematical point of view?  Would it be the same as writing that the equality has to hold for all states rather than only the ones visited by pi?**
>
> That is right! The equality should hold for all states in $s_t$'s equivalence class under $\Phi$ (i.e., all states that share the same value for the state variables that are "selected” by $\Phi$).
>
> > **About "Narrow trajectory distributions". If the environment transitons leads to a diverse set of states, would we still observe policy confounding?**
>
> That depends on how diverse the set of states is and how frequently each state is visited. In general, if the environment is random enough such that the agent visits all states sufficiently frequently, we would most likely avoid policy confounding. The same goes for the policy. If the policy is stochastic enough such that the agent visits all states sufficiently frequently, the issue may be avoided. So, yes, exploration does help! Both domain randomization and exploration are discussed in Section 6.
>
> > **The paper makes a link between policy confounding and causality (or lack of it). Are there any tools from causality that could be used to address the problem?**
>
> Not directly as far as we know. The question of how tools from causal inference can be adapted to prevent policy confounding will be investigated in future work. However, one promising option would be to exploit the notion of prediction invariance (Peters et al., 2016; Arjovsky et al., 2019) since state representations that are robust to trajectory deviations are those that are invariant across policies.
>
> > **Is policy confounding inevitable to some extent? It seems to be a consequence of having policies focus on certain parts of the state space.**
>
> Good point! The way the RL objective is normally formulated makes avoiding policy confounding challenging. However, we don't think the issue is inevitable. We believe that, apart from maximizing reward, the agent should strive for learning representations grounded in causal relationships rather than mere statistical associations. Incorporating auxiliary objectives may help in this regard. However, this is again beyond the scope of this paper. Our goal in this paper was to highlight the phenomenon of policy confounding and raise awareness about its risks.

---

> ### Author Response · Authors · 2023-11-22
>
> Dear reviewer,
>
> As the discussion period comes to a close, we kindly request your feedback regarding our rebuttal. Have the responses to your questions effectively addressed the concerns you raised?

---

> > ### Comment · Reviewer_BRdu · 2023-11-22
> >
> > Thank you for the detailed responses.
> > Having read the other reviews, it does occur to me that certain parts could be explained more clearly.
> > I think the contribution, identifying "policy confounding" as a problem, is quite interesting. If the paper had one more contribution, say, some causality-inspired solution (even if it's not scalable) or demonstrating the effect in less toyish environments, then it would be a clear accept for me but in its present form, I will keep my current score.

---

### Author Response · Authors · 2023-11-20

Dear reviewers,

As the discussion period comes to a close, we kindly request your feedback regarding our rebuttal. We value your input on whether our response effectively addressed your concerns and if there are any remaining questions or points you would like us to elaborate on for further clarity.

---

### Meta-Review · Area_Chair_fAZi · 2023-12-05

**Metareview:**

This work theoretically introduces the "policy confounding" problem, whereby agents pick up on habits that work well in specific domains/tasks but do not generalize. This is grounded in the real world and relates to spurious correlations in the behavior space. The paper contains theory which some of the reviewers found confusing, while the experiments were described as contrived and potentially designed to demonstrate the problem rather than demonstrate the utility of the formalism to solve problems actively being considered by the community. Overall I agree with the majority of the reviewers that the contribution seems light, and a re-write to focus on relevant machine learning problems from the get-go would make the paper much stronger for future submission.

**Justification For Why Not Higher Score:**

The experiments are toy even for RL standards and this then acts more like a theory or position paper. In that case, it would be better if this is more directly linked to some relevant problems we are currently considering in the ML community.

**Justification For Why Not Lower Score:**

N/A

---

### Decision · Program_Chairs · 2024-01-16

Reject